# Non-stationary Transformers:
# Exploring the Stationarity in Time Series Forecasting

**Yong Liu,\* Haixu Wu,\* Jianmin Wang, Mingsheng Long**✉
School of Software, BNRist, Tsinghua University, China
{liuyong21,whx20}@mails.tsinghua.edu.cn, {jimwang,mingsheng}@tsinghua.edu.cn

## Abstract

Transformers have shown great power in time series forecasting due to their global-range modeling ability. However, their performance can degenerate terribly on non-stationary real-world data in which the joint distribution changes over time. Previous studies primarily adopt stationarization to attenuate the non-stationarity of original series for better predictability. But the stationarized series deprived of inherent non-stationarity can be less instructive for real-world bursty events forecasting. This problem, termed *over-stationarization* in this paper, leads Transformers to generate indistinguishable temporal attentions for different series and impedes the predictive capability of deep models. To tackle the dilemma between series predictability and model capability, we propose *Non-stationary Transformers* as a generic framework with two interdependent modules: Series Stationarization and De-stationary Attention. Concretely, Series Stationarization unifies the statistics of each input and converts the output with restored statistics for better predictability. To address the over-stationarization problem, De-stationary Attention is devised to recover the intrinsic non-stationary information into temporal dependencies by approximating distinguishable attentions learned from raw series. Our Non-stationary Transformers framework consistently boosts mainstream Transformers by a large margin, which reduces MSE by 49.43% on Transformer, 47.34% on Informer, and 46.89% on Reformer, making them the state-of-the-art in time series forecasting. Code is available at this repository: https://github.com/thuml/Nonstationary_Transformers.

## 1 Introduction

Time series forecasting has become increasingly ubiquitous in real-world applications, such as weather forecasting, energy consumption planning, and financial risk assessment. Recently, Transformers [32] have achieved progressive breakthrough on extensive areas [11, 12, 10, 22]. Especially in time series forecasting, credited to their stacked structure and the capability of attention mechanisms, Transformers can naturally capture the temporal dependencies from deep multi-level features [37, 17, 20, 35], thereby fitting the series forecasting task perfectly.

Despite the remarkable architectural design, it is still challenging for Transformers to predict real-world time series because of the non-stationarity of data. Non-stationary time series is characterized by the continuous change of statistical properties and joint distribution over time, which makes the time series less predictable [6, 14]. Besides, it is a fundamental problem to make deep models generalize well on a varying distribution [26, 19, 5]. In previous work, it is generally acknowledged to pre-process the time series by stationarization [24, 27, 15], which can attenuate the non-stationarity of raw time series for better predictability and provide more stable data distribution for deep models.

---

\*Equal Contribution

36th Conference on Neural Information Processing Systems (NeurIPS 2022).

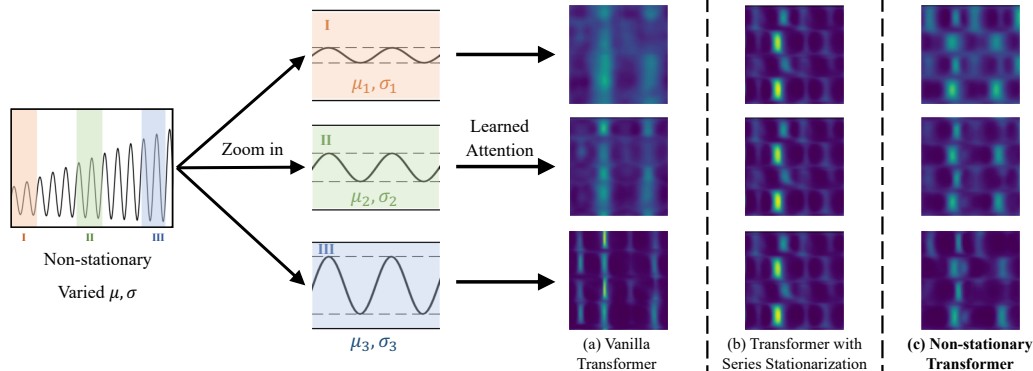

Figure 1: Visualization of learned temporal attentions for different series with varied mean $\mu$ and standard deviation $\sigma$. (a) is from the vanilla Transformer [32] trained on raw series. (b) is from the Transformer trained on stationarized series, which presents similar attentions. (c) is from Non-stationary Transformers, which involves De-stationary Attention to avoid over-stationarization.

However, non-stationarity is the inherent property of real-world time series and also good guidance for discovering temporal dependencies for forecasting. Experimentally, we observe that training on the stationarized series will undermine the distinction of attentions learned by Transformers. While vanilla Transformers [32] can capture distinct temporal dependencies from different series in Figure 1(a), Transformers trained on the stationarized series tend to generate indistinguishable attentions in Figure 1(b). This problem, named by the *over-stationarization*, will bring unexpected side-effect that makes Transformers fail to capture eventful temporal dependencies, limit the model's predictive ability, and even induce the model to generate outputs with huge non-stationarity deviation from the ground truth. Thus, *how to attenuate time series non-stationarity towards better predictability and mitigate the over-stationarization problem for model capability simultaneously* is the key problem to further improve the performance of forecasting.

In this paper, we explore the effect of stationarization in time series forecasting and propose *Non-stationary Transformers* as a general framework, which empowers Transformer [32] and its efficient variants [17, 37, 35] with great predictive ability for real-world time series. The proposed framework involves two interdependent modules: Series Stationarization to increase the predictability of non-stationary series and De-stationary Attention to alleviate over-stationarization. Technically, Series Stationarization adopts a simple but effective normalization strategy to unify the key statistics of each series without extra parameters. And De-stationary Attention approximates the attention of unstationarized data and compensates the intrinsic non-stationarity of raw series. Benefiting from the above designs, Non-stationary Transformers can take advantage of the great predictability of stationarized series and crucial temporal dependencies discovered from original non-stationary data. Our method achieves state-of-the-art performance on six real-world benchmarks and can generalize to various Transformers for further improvement. The contributions lie in three folds:

- We refine that the predictive capability of non-stationary series is essential in real-world forecasting. By detailed analysis, we find out that current stationarization approaches will lead to the over-stationarization problem, limiting the predictive capability of Transformers.

- We propose Non-stationary Transformers as a generic framework, including Series Stationarization to make the series more predictable and De-stationary Attention to avoid the over-stationarization problem by re-incorporating the non-stationarity of original series.

- Non-stationary Transformers consistently boosts four mainstream Transformers by a large margin and achieves state-of-the-art performance on six real-world benchmarks.

## 2 Related Work

### 2.1 Deep Models for Time Series Forecasting

In recent years, deep models with elaboratively designed architectures have achieved great progress in time series forecasting. RNN-based models [33, 36, 23, 29, 30] are proposed for application in an

autoregressive manner for sequence modeling, but the recurrent structure can suffer from modeling long-term dependency. Soon afterward, Transformer [32] emerges and shows great power in sequence modeling. To overcome the quadratic computation growth on sequence length, subsequent works aim to reduce Self-Attention's complexity. Especially in time series forecasting, Informer [37] extends Self-Attention with KL-divergence criterion to select dominant queries. Reformer [17] introduces local-sensitive hashing (LSH) to approximate attention by allocated similar queries. Not only improved by reduced complexity, the following models further develop delicate building blocks for time series forecasting. Autoformer [35] fuses the decomposition blocks into a canonical structure and develops Auto-Correlation to discover series-wise connections. Pyraformer [21] designs pyramid attention module (PAM) to capture temporal dependencies with different hierarchies. Other deep but Transformer-free models also achieve remarkable performance. N-BEATS [25] proposes the explicit decomposition of trend and seasonal terms with strong interpretability. N-HiTS [9] introduces hierarchical layout and multi-rate sampling for tackling time series with respective frequency bands. In this paper, different from previous works focusing on architectural design, we analyze the series forecasting task from the basic view of stationarity, which is an essential property of time series [6, 14]. It is also notable that as a general framework, our proposed Non-stationary Transformers can be easily applied to various Transformer-based models.

## 2.2   Stationarization for Time Series Forecasting

While stationarity is important to the predictability of time series [6, 14], real-world series always present non-stationarity. To tackle this problem, the classical statistical method ARIMA [7, 8] stationarizes the time series through differencing. As for deep models, since the distribution-varying problem accompanied by non-stationarity makes deep forecasting even more intractable, stationarization methods are widely explored and always adopted as the pre-processing for deep model inputs. Adaptive Norm [24] applies z-score normalization for each series fragment by global statistics of a sampled set. DAIN [27] employs a nonlinear neural network to adaptively stationarize time series with observed training distribution. RevIN [15] introduces a two-stage instance normalization [31] that transforms model input and output respectively to reduce the discrepancy of each series. In contrast, we find out that directly stationarizing time series will damage the model's capability of modeling specific temporal dependency. Therefore, unlike previous methods, in addition to the stationarization, Non-stationary Transformers further develops De-stationary Attention to bring the intrinsic non-stationarity of the raw series back to attention.

## 3   Non-stationary Transformers

As aforementioned, stationarity is an important element of time series predictability. Previous "direct stationarization" designs can attenuate non-stationarity of series for better predictability, but they obviously neglect inherent properties of real-world series, which will result in the over-stationarization problem as stated in Figure 1. To deal with the dilemma, we go beyond previous works and propose *Non-stationary Transformers* as a generic framework. Our model involves two complementary parts: Series Stationarization to attenuate time series non-stationarity and De-stationary Attention to re-incorporate non-stationary information of raw series. Empowered by these designs, Non-stationary Transformers can improve data predictability and maintain model capability simultaneously.

### 3.1   Series Stationarization

Non-stationary time series make the forecasting task intractable for deep models because it is hard for them to generalize well on series with changed statistics during inference, typically varied mean and standard deviation. The pilot work, RevIN [15] applies instance normalization with learnable affine parameters to each input and restores the statistics to the corresponding output, which makes each series follow a similar distribution. Experimentally, we find that this design also works well without learnable parameters. Thus, we propose a more straightforward but effective design to wrap Transformers as the base model without extra parameters, naming by Series Stationarization. As is shown in Figure 2, it contains two corresponding operations: Normalization module at first to deal with the non-stationary series caused by varied mean and standard deviation, and De-normalization module at the end to transform the model outputs back with original statistics. Here are the details.

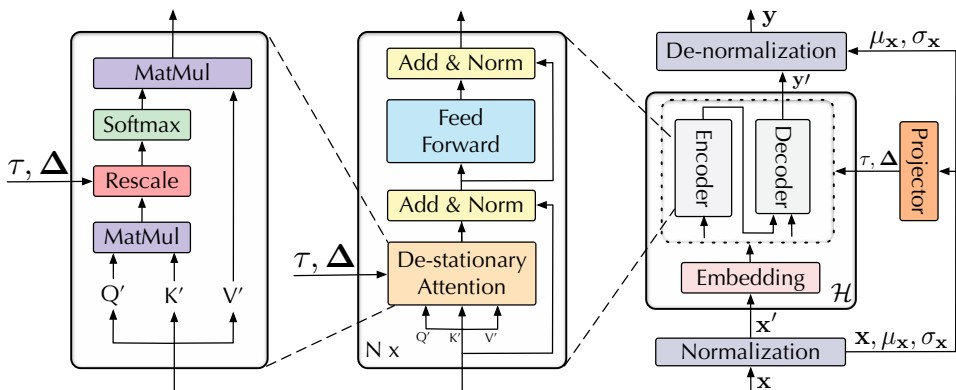

Figure 2: Non-stationary Transformers. Series Stationarization is adopted as a wrapper on the base model to normalize each incoming series and de-normalize the output. De-stationary Attention replaces the original Attention mechanism to approximate attention learned from unstationarized series, which rescales current temporal dependency weights with learned de-stationary factors $\tau, \boldsymbol{\Delta}$.

**Normalization module**   To attenuate the non-stationarity of each input series, we conduct normalization on the temporal dimension by a sliding window over time. For each input series $\mathbf{x} = [x_1, x_2, ..., x_S]^\top \in \mathbb{R}^{S \times C}$, we transform it by translation and scaling operations and obtain $\mathbf{x}' = [x_1', x_2', ..., x_S']^\top \in \mathbb{R}^{S \times C}$, where $S$ and $C$ denote the sequence length and variable number respectively. The Normalization module can be formulated as follows:

$$\mu_{\mathbf{x}} = \frac{1}{S} \sum_{i=1}^{S} x_i, \ \sigma_{\mathbf{x}}^2 = \frac{1}{S} \sum_{i=1}^{S} (x_i - \mu_{\mathbf{x}})^2, \ x_i' = \frac{1}{\sigma_{\mathbf{x}}} \odot (x_i - \mu_{\mathbf{x}}), \tag{1}$$

where $\mu_{\mathbf{x}}, \sigma_{\mathbf{x}} \in \mathbb{R}^{C \times 1}$, $\frac{1}{\sigma_{\mathbf{x}}}$ means the element-wise division and $\odot$ is the element-wise product. Note that Normalization module decreases the distributional discrepancy among each input time series, making the distribution of the model input more stable.

**De-normalization module**   As shown in Figure 2, after the base model $\mathcal{H}$ predicting the future value with length-$O$, we adopt De-normalization to transform the model output $\mathbf{y}' = [y_1', y_2', ..., y_O']^\top \in \mathbb{R}^{O \times C}$ with $\sigma_{\mathbf{x}}$ and $\mu_{\mathbf{x}}$ and obtain $\hat{\mathbf{y}} = [\hat{y}_1, \hat{y}_2, ..., \hat{y}_O]^\top$ as the eventual forecasting results. The De-normalization module can be formulated as follows:

$$\mathbf{y}' = \mathcal{H}(\mathbf{x}'), \ \hat{y}_i = \sigma_{\mathbf{x}} \odot (y_i' + \mu_{\mathbf{x}}). \tag{2}$$

By means of the two-stage transformation, the base models will receive stationarized inputs, which follow a stable distribution and are easier to generalize. This design also makes the model equivariant to translational and scaling perturbance of time series, thereby benefiting real-world series forecasting.

### 3.2  De-stationary Attention

While the statistics of each time series are explicitly restored to the corresponding prediction, the non-stationarity of the original series cannot be fully recovered only by De-normalization. For instance, Series Stationarization can generate the same stationarized input $\mathbf{x}'$ from distinct time series $\mathbf{x}_1, \mathbf{x}_2$ (i.e. $\mathbf{x}_2 = \alpha \mathbf{x}_1 + \beta$), and the base model will get identical attention that fails to capture crucial temporal dependencies entangled with non-stationarity (Figure 1). In other words, the undermined effects caused by over-stationarization happen inside the deep model, especially in the calculation of attention. Furthermore, non-stationary time series are fragmented and normalized into several series chunks with the same mean and variance, which follow more similar distributions than the raw data before stationarization. Thus, the model is more likely to generate over-stationary and uneventful outputs, which is irreconcilable with the natural non-stationarity of the original series.

To tackle the over-stationarization problem caused by Series Stationarization, we propose a novel De-stationary Attention mechanism, which can approximate the attention that is obtained without stationarization and discover the particular temporal dependencies from original non-stationary data.

**Analysis of the plain model**  As mentioned above, the over-stationarization problem is caused by the vanishment of inherent non-stationarity information, which will make the base model fail to capture eventful temporal dependencies for forecasting. Therefore, we try to approximate the attention learned from the original non-stationary series. We start from the formula of Self-Attention [32]:

$$\text{Attn}(\mathbf{Q}, \mathbf{K}, \mathbf{V}) = \text{Softmax}\left(\frac{\mathbf{Q}\mathbf{K}^\top}{\sqrt{d_k}}\right)\mathbf{V}, \tag{3}$$

where $\mathbf{Q}, \mathbf{K}, \mathbf{V} \in \mathbb{R}^{S \times d_k}$ are length-$S$ queries, keys and values of $d_k$-dimension respectively, and $\text{Softmax}(\cdot)$ is conducted row by row. To simplify the analysis, we assume the embedding and feed-forward layers $f$ to hold the linear properties[2] and $f$ is conducted separately on each time point, that is, each query token in $\mathbf{Q} = [q_1, q_2, ..., q_S]^\top$ can be calculated as $q_i = f(x_i)$ with respect to the input series $\mathbf{x} = [x_1, x_2, \cdots, x_S]^\top$. Since it is a convention to conduct normalization on each time series variable to avoid certain variable that dominates the scale, we can further assume each variable of series $\mathbf{x}$ shares the same variance, and thus original $\sigma_\mathbf{x} \in \mathbb{R}^{C \times 1}$ is reduced to a scalar. After Normalization module, the model receives the stationarized input $\mathbf{x}' = (\mathbf{x} - \mathbf{1}\mu_\mathbf{x}^\top)/\sigma_\mathbf{x}$, where $\mathbf{1} \in \mathbb{R}^{S \times 1}$ is an all-ones vector. Based on the linear property assumption, it can be proved that the Attention layer will receive $\mathbf{Q}' = [f(x_1'), ..., f(x_S')]^\top = (\mathbf{Q} - \mathbf{1}\mu_\mathbf{Q}^\top)/\sigma_\mathbf{x}$, where $\mu_\mathbf{Q} \in \mathbb{R}^{d_k \times 1}$ is the mean of $\mathbf{Q}$ along the temporal dimension (See Appendix for a detailed proof). And so is the corresponding transformed $\mathbf{K}', \mathbf{V}'$. Without Series Stationarization, the input of $\text{Softmax}(\cdot)$ in Self-Attention should be $\mathbf{Q}\mathbf{K}^\top/\sqrt{d_k}$, while now the attention is calculated based on $\mathbf{Q}', \mathbf{K}'$:

$$\mathbf{Q}'\mathbf{K}'^\top = \frac{1}{\sigma_\mathbf{x}^2}\left(\mathbf{Q}\mathbf{K}^\top - \mathbf{1}(\mu_\mathbf{Q}^\top\mathbf{K}^\top) - (\mathbf{Q}\mu_\mathbf{K})\mathbf{1}^\top + \mathbf{1}(\mu_\mathbf{Q}^\top\mu_\mathbf{K})\mathbf{1}^\top\right),$$

$$\text{Softmax}\left(\frac{\mathbf{Q}\mathbf{K}^\top}{\sqrt{d_k}}\right) = \text{Softmax}\left(\frac{\sigma_\mathbf{x}^2\,\mathbf{Q}'\mathbf{K}'^\top + \mathbf{1}(\mu_\mathbf{Q}^\top\mathbf{K}^\top) + (\mathbf{Q}\mu_\mathbf{K})\mathbf{1}^\top - \mathbf{1}(\mu_\mathbf{Q}^\top\mu_\mathbf{K})\mathbf{1}^\top}{\sqrt{d_k}}\right). \tag{4}$$

We find that $\mathbf{Q}\mu_\mathbf{K} \in \mathbb{R}^{S \times 1}$ and $\mu_\mathbf{Q}^\top\mu_\mathbf{K} \in \mathbb{R}$, and they are repeatedly operated on each column and element of $\sigma_\mathbf{x}^2\mathbf{Q}'\mathbf{K}'^\top \in \mathbb{R}^{S \times S}$ respectively. Since $\text{Softmax}(\cdot)$ is invariant to the same translation on the row dimension of input, we have the following equation:

$$\text{Softmax}\left(\frac{\mathbf{Q}\mathbf{K}^\top}{\sqrt{d_k}}\right) = \text{Softmax}\left(\frac{\sigma_\mathbf{x}^2\,\mathbf{Q}'\mathbf{K}'^\top + \mathbf{1}\mu_\mathbf{Q}^\top\mathbf{K}^\top}{\sqrt{d_k}}\right). \tag{5}$$

Equation 5 deduces a direct expression of the attention $\text{Softmax}\left(\mathbf{Q}\mathbf{K}^\top/\sqrt{d_k}\right)$ learned from raw series $\mathbf{x}$. Except for the current $\mathbf{Q}', \mathbf{K}'$ from stationarized series $\mathbf{x}'$, this expression also requires the non-stationary information $\sigma_\mathbf{x}, \mu_\mathbf{Q}, \mathbf{K}$ that are eliminated by Series Stationarization.

**De-stationary Attention**  To recover the original attention on non-stationary series, we attempt to bring the vanished non-stationary information back to its calculation. Based on Equation 5, the key is to approximate the positive scaling scalar $\tau = \sigma_\mathbf{x}^2 \in \mathbb{R}^+$ and shifting vector $\mathbf{\Delta} = \mathbf{K}\mu_\mathbf{Q} \in \mathbb{R}^{S \times 1}$, which are defined as *de-stationary factors*. Since the strict linear property hardly holds for a deep model, other than estimating and utilizing real factors with great effort, we try to learn de-stationary factors directly from the statistics of unstationarized $\mathbf{x}, \mathbf{Q}$ and $\mathbf{K}$ by a simple but effective multi-layer perceptron layer. As we can only discover limited non-stationary information from current $\mathbf{Q}', \mathbf{K}'$, the unique and reasonable source to compensate non-stationarity is the original $\mathbf{x}$ without being normalized. Thus, as a direct deep learning implementation of Equation 5, we apply a multi-layer perceptron as the projector to learn de-stationary factors $\tau, \mathbf{\Delta}$ from the statistics $\mu_\mathbf{x}, \sigma_\mathbf{x}$ of unstationarized $\mathbf{x}$ individually. And the De-stationary Attention is calculated as follows:

$$\log \tau = \text{MLP}(\sigma_\mathbf{x}, \mathbf{x}), \quad \mathbf{\Delta} = \text{MLP}(\mu_\mathbf{x}, \mathbf{x}),$$

$$\text{Attn}(\mathbf{Q}', \mathbf{K}', \mathbf{V}', \tau, \mathbf{\Delta}) = \text{Softmax}\left(\frac{\tau\,\mathbf{Q}'\mathbf{K}'^\top + \mathbf{1}\mathbf{\Delta}^\top}{\sqrt{d_k}}\right)\mathbf{V}', \tag{6}$$

where the de-stationary factors $\tau$ and $\mathbf{\Delta}$ are shared by De-stationary Attention of all layers (Figure 2). De-stationary Attention mechanism learns the temporal dependencies from both stationarized series

---

[2]Function $f$ has the linear property if it satisfies that $f(ax + by) = af(x) + bf(y)$, where $a, b$ are scalar constants and $x, y$ are vector variables.

$\mathbf{Q}'$, $\mathbf{K}'$ and non-stationary series $\mathbf{x}$, $\mu_{\mathbf{x}}$, $\sigma_{\mathbf{x}}$, and multiplies by the stationarized values $\mathbf{V}'$. Therefore, it can benefit from the predictability of stationarized series and maintain the inherent temporal dependencies of raw series simultaneously.

**Overall architecture**   Following the prior use of Transformers [37, 35] in time series forecasting, we adopt the standard Encoder-Decoder structure (Figure 2), where the encoder is to extract information from past observations, and the decoder is to aggregate past information and refine the prediction from simple initialization. The canonical Non-stationary Transformer is wrapped by Series Stationarization to both the input and output of vanilla Transformer [32], and replacing the Self-Attention by our proposed De-stationary Attention, which can boost the non-stationary series predictive capability of the base model. For the Transformer variants [17, 37, 35], we transform the terms inside $\mathrm{Softmax}(\cdot)$ with the de-stationary factors $\tau$, $\boldsymbol{\Delta}$ to re-integrate the non-stationary information (See Appendix for the implementation details).

## 4   Experiments

We conduct extensive experiments to evaluate the performance of Non-stationary Transformers on six real-world time series forecasting benchmarks and further validate the generality of the proposed framework on various mainstream Transformer variants.

**Datasets**   Here are the descriptions of the datasets: (1) **Electricity** [3] records the hourly electricity consumption of 321 clients from 2012 to 2014. (2) **ETT** [37] contains the time series of oil de-stationary factors and power load collected by electricity transformers from July 2016 to July 2018. ETTm1 /ETTm2 are recorded every 15 minutes, and ETTh1/ETTh2 are recorded every hour. (3) **Exchange** [18] collects the panel data of daily exchange rates from 8 countries from 1990 to 2016. (4) **ILI** [1] collects the ratio of influenza-like illness patients versus the total patients in one week, which is reported weekly by Centers for Disease Control and Prevention of the United States from 2002 and 2021. (5) **Traffic** [2] contains hourly road occupancy rates measured by 862 sensors on San Francisco Bay area freeways from January 2015 to December 2016. (6) **Weather** [4] includes meteorological time series with 21 weather indicators collected every 10 minutes from the Weather Station of the Max Planck Biogeochemistry Institute in 2020.

Especially, in this paper, we adopt the Augmented Dick-Fuller (ADF) test statistic [13] as the metric to quantitatively measure the *degree of stationarity*. A smaller ADF test statistic indicates a higher degree of stationarity, which means the distribution is more stable. Table 1 summarizes the overall statistics of the datasets and lists them in ascending order by degree of stationarity. We follow the standard protocol that divides each dataset into the training, validation, and testing subsets according to the chronological order. The split ratio is 6:2:2 for the ETT dataset and 7:1:2 for others.

Table 1: Summary of datasets. Smaller ADF test statistic indicates more stationary dataset.

| Dataset | Variable Number | Sampling Frequency | Total Observations | ADF Test Statistic |
|---|---|---|---|---|
| Exchange | 8 | 1 Day | 7,588 | -1.889 |
| ILI | 7 | 1 Week | 966 | -5.406 |
| ETTm2 | 7 | 15 Minutes | 69,680 | -6.225 |
| Electricity | 321 | 1 Hour | 26,304 | -8.483 |
| Traffic | 862 | 1 Hour | 17,544 | -15.046 |
| Weather | 21 | 10 Minutes | 52,695 | -26.661 |

**Baselines**   We evaluate the vanilla Transformer [32] equipped by the Non-stationary Transformers framework in both multivariate and univariate settings to demonstrate its effectiveness. For multivariate forecasting, we include six state-of-the-art deep forecasting models: Autoformer [35], Pyraformer [21], Informer [37], LogTrans [20], Reformer [17] and LSTNet [18]. For univariate forecasting, we include seven competitive baselines: N-HiTS [9], N-BEATS [25], Autoformer [35], Pyraformer [21], Informer [37], Reformer [17] and ARIMA [7]. In addition, we adopt the proposed framework on both the canonical and efficient variants of Transformers: Transformer [32], Informer [37], Reformer [17] and Autoformer [35] to validate the generality of our framework.

**Implementation details**   All the experiments are implemented with PyTorch [28] and conducted on a single NVIDIA TITAN V 12GB GPU. Each model is trained by ADAM [16] using L2 loss with

the initial learning rate of $10^{-4}$ and batch size of 32. Each Transformer-based model contains two encoder layers and one decoder layer. Considering the efficiency of hyperparameters search, we use two-layer perceptron projector with the hidden dimension varying in $\{64, 128, 256\}$ in De-stationary Attention. We repeat each experiment three times with different random seeds and report the test MSE/MAE under different prediction lengths, and the standard deviations are also provided in Appendix. A lower MSE/MAE indicates better performance.

## 4.1 Main Results

**Forecasting results**  As for multivariate forecasting results, the vanilla Transformer equipped with our framework consistently achieves state-of-the-art performance in all benchmarks and prediction lengths (Table 2). Notably, Non-stationary Transformer outperforms other deep models impressively on datasets characterized by high non-stationarity: under the prediction length of 336, we achieve **17%** MSE reduction ($0.509 \rightarrow 0.421$) on Exchange and **25%** ($2.669 \rightarrow 2.010$) on ILI compared to previous state-of-the-art results, which indicates that the potential of deep model is still constrained on non-stationary data. We also list the univariate results of two typical datasets with different stationarity in Table 3. Non-stationary Transformer still realizes remarkable forecasting performance.

Table 2: Forecasting results comparison under different prediction lengths $O \in \{96, 192, 336, 720\}$. The input sequence length is set to 36 for ILI and 96 for the others. Additional results (ETTm1, ETTh1, ETTh2) can be found in Appendix.

| Models | | **Ours** | | Autoformer [35] | | Pyraformer [21] | | Informer [37] | | LogTrans [20] | | Reformer [17] | | LSTNet [18] | |
|---|---|---|---|---|---|---|---|---|---|---|---|---|---|---|---|
| Metric | | MSE | MAE | MSE | MAE | MSE | MAE | MSE | MAE | MSE | MAE | MSE | MAE | MSE | MAE |
| Exchange | 96 | **0.111** | **0.237** | 0.197 | 0.323 | 0.852 | 0.780 | 0.847 | 0.752 | 0.968 | 0.812 | 1.065 | 0.829 | 1.551 | 1.058 |
| | 192 | **0.219** | **0.335** | 0.300 | 0.369 | 0.993 | 0.858 | 1.204 | 0.895 | 1.040 | 0.851 | 1.188 | 0.906 | 1.477 | 1.028 |
| | 336 | **0.421** | **0.476** | 0.509 | 0.524 | 1.240 | 0.958 | 1.672 | 1.036 | 1.659 | 1.081 | 1.357 | 0.976 | 1.507 | 1.031 |
| | 720 | **1.092** | **0.769** | 1.447 | 0.941 | 1.711 | 1.093 | 2.478 | 1.310 | 1.941 | 1.127 | 1.510 | 1.016 | 2.285 | 1.243 |
| ILI | 24 | **2.294** | **0.945** | 3.483 | 1.287 | 5.800 | 1.693 | 5.764 | 1.677 | 4.480 | 1.444 | 4.400 | 1.382 | 6.026 | 1.770 |
| | 36 | **1.825** | **0.848** | 3.103 | 1.148 | 6.043 | 1.733 | 4.755 | 1.467 | 4.799 | 1.467 | 4.783 | 1.448 | 5.340 | 1.668 |
| | 48 | **2.010** | **0.900** | 2.669 | 1.085 | 6.213 | 1.763 | 4.763 | 1.469 | 4.800 | 1.468 | 4.832 | 1.465 | 6.080 | 1.787 |
| | 60 | **2.178** | **0.963** | 2.770 | 1.125 | 6.531 | 1.814 | 5.264 | 1.564 | 5.278 | 1.560 | 4.882 | 1.483 | 5.548 | 1.720 |
| ETTm2 | 96 | **0.192** | **0.274** | 0.255 | 0.339 | 0.409 | 0.488 | 0.365 | 0.453 | 0.768 | 0.642 | 0.658 | 0.619 | 3.142 | 1.365 |
| | 192 | **0.280** | **0.339** | 0.281 | 0.340 | 0.673 | 0.641 | 0.533 | 0.563 | 0.989 | 0.757 | 1.078 | 0.827 | 3.154 | 1.369 |
| | 336 | **0.334** | **0.361** | 0.339 | 0.372 | 1.210 | 0.846 | 1.363 | 0.887 | 1.334 | 0.872 | 1.549 | 0.972 | 3.160 | 1.369 |
| | 720 | **0.417** | **0.413** | 0.422 | 0.419 | 4.044 | 1.526 | 3.379 | 1.388 | 3.048 | 1.328 | 2.631 | 1.242 | 3.171 | 1.368 |
| Electricity | 96 | **0.169** | **0.273** | 0.201 | 0.317 | 0.498 | 0.299 | 0.274 | 0.368 | 0.258 | 0.357 | 0.312 | 0.402 | 0.680 | 0.645 |
| | 192 | **0.182** | **0.286** | 0.222 | 0.334 | 0.828 | 0.312 | 0.296 | 0.386 | 0.266 | 0.368 | 0.348 | 0.433 | 0.725 | 0.676 |
| | 336 | **0.200** | **0.304** | 0.231 | 0.338 | 1.476 | 0.326 | 0.300 | 0.394 | 0.280 | 0.380 | 0.350 | 0.433 | 0.828 | 0.727 |
| | 720 | **0.222** | **0.321** | 0.254 | 0.361 | 4.090 | 0.372 | 0.373 | 0.439 | 0.283 | 0.376 | 0.340 | 0.420 | 0.957 | 0.811 |
| Traffic | 96 | **0.612** | **0.338** | 0.613 | 0.388 | 0.684 | 0.393 | 0.719 | 0.391 | 0.684 | 0.384 | 0.732 | 0.423 | 1.107 | 0.685 |
| | 192 | **0.613** | **0.340** | 0.616 | 0.382 | 0.692 | 0.394 | 0.696 | 0.379 | 0.685 | 0.390 | 0.733 | 0.420 | 1.157 | 0.706 |
| | 336 | **0.618** | **0.328** | 0.622 | 0.337 | 0.699 | 0.396 | 0.777 | 0.420 | 0.733 | 0.408 | 0.742 | 0.420 | 1.216 | 0.730 |
| | 720 | **0.653** | **0.355** | 0.660 | 0.408 | 0.712 | 0.404 | 0.864 | 0.472 | 0.717 | 0.396 | 0.755 | 0.423 | 1.481 | 0.805 |
| Weather | 96 | **0.173** | **0.223** | 0.266 | 0.336 | 0.354 | 0.392 | 0.300 | 0.384 | 0.458 | 0.490 | 0.689 | 0.596 | 0.594 | 0.587 |
| | 192 | **0.245** | **0.285** | 0.307 | 0.367 | 0.673 | 0.597 | 0.598 | 0.544 | 0.658 | 0.589 | 0.752 | 0.638 | 0.560 | 0.565 |
| | 336 | **0.321** | **0.338** | 0.359 | 0.395 | 0.634 | 0.592 | 0.578 | 0.523 | 0.797 | 0.652 | 0.639 | 0.596 | 0.597 | 0.587 |
| | 720 | **0.414** | **0.410** | 0.419 | 0.428 | 0.942 | 0.723 | 1.059 | 0.741 | 0.869 | 0.675 | 1.130 | 0.792 | 0.618 | 0.599 |

**Framework generality**  We apply our framework to four mainstream Transformers and report the performance promotion of each model (Table 4). Our method consistently improves the forecasting ability of different models. Overall, it achieves averaged **49.43%** promotion on Transformer, **47.34%** on Informer, **46.89%** on Reformer and **10.57%** on Autoformer, making each of them surpass previous state-of-the-art. Compared to native blocks of the models, there is hardly any parameter and computation increase by applying our framework (See Appendix for details), and thereby their computational complexities can be preserved. It validates that Non-stationary Transformer is an effective and lightweight framework that can be widely applied to Transformer-based models and enhances their non-stationary predictability to achieve state-of-the-art performance.

Table 3: Univariate results under different prediction lengths $O \in \{96, 192, 336, 720\}$ on two typical datasets with strong non-stationary. The input sequence length is set to 96.

| Models | Ours | | N-HiTS [9] | | N-BEATS [25] | | Autoformer [35] | | Pyraformer [21] | | Informer [37] | | Reformer [17] | | ARIMA [6] | |
|---|---|---|---|---|---|---|---|---|---|---|---|---|---|---|---|---|
| Metric | MSE | MAE | MSE | MAE | MSE | MAE | MSE | MAE | MSE | MAE | MSE | MAE | MSE | MAE | MSE | MAE |
| Exchange 96 | **0.104** | **0.235** | 0.114 | 0.248 | 0.156 | 0.299 | 0.241 | 0.387 | 0.290 | 0.439 | 0.591 | 0.615 | 1.327 | 0.944 | 0.112 | 0.245 |
| Exchange 192 | **0.230** | **0.375** | 0.250 | 0.387 | 0.669 | 0.665 | 0.273 | 0.403 | 0.594 | 0.644 | 1.183 | 0.912 | 1.258 | 0.924 | 0.304 | 0.404 |
| Exchange 336 | **0.432** | **0.509** | 0.434 | 0.516 | 0.611 | 0.605 | 0.508 | 0.539 | 0.962 | 0.824 | 1.367 | 0.984 | 2.179 | 1.296 | 0.736 | 0.598 |
| Exchange 720 | **0.782** | **0.682** | 1.061 | 0.773 | 1.111 | 0.860 | 0.991 | 0.768 | 1.285 | 0.958 | 1.872 | 1.072 | 1.280 | 0.953 | 1.871 | 0.935 |
| ETTm2 96 | 0.069 | 0.193 | 0.092 | 0.232 | 0.082 | 0.219 | **0.065** | **0.189** | 0.074 | 0.208 | 0.088 | 0.225 | 0.131 | 0.288 | 0.211 | 0.362 |
| ETTm2 192 | **0.109** | **0.249** | 0.128 | 0.276 | 0.120 | 0.268 | 0.118 | 0.256 | 0.116 | 0.252 | 0.132 | 0.283 | 0.186 | 0.354 | 0.261 | 0.406 |
| ETTm2 336 | **0.139** | **0.286** | 0.165 | 0.314 | 0.226 | 0.370 | 0.154 | 0.305 | 0.143 | 0.295 | 0.180 | 0.336 | 0.220 | 0.381 | 0.317 | 0.448 |
| ETTm2 720 | **0.180** | **0.331** | 0.243 | 0.397 | 0.188 | 0.338 | 0.182 | 0.335 | 0.197 | 0.338 | 0.300 | 0.435 | 0.267 | 0.430 | 0.366 | 0.487 |

Table 4: Performance promotion by applying the proposed framework to Transformer and its variants. We report the averaged MSE/MAE of all prediction lengths (stated in Table 2) and the relative MSE reduction ratios (Promotion) by our framework. Full results (under all prediction lengths and promotion on ETSformer [34], FEDformer [38]) can be found in Appendix.

| Dataset | Exchange | | ILI | | ETTm2 | | Electricity | | Traffic | | Weather | |
|---|---|---|---|---|---|---|---|---|---|---|---|---|
| Model | MSE | MAE | MSE | MAE | MSE | MAE | MSE | MAE | MSE | MAE | MSE | MAE |
| Transformer | 1.425 | 0.915 | 4.864 | 1.460 | 1.501 | 0.869 | 0.277 | 0.372 | 0.665 | 0.363 | 0.657 | 0.573 |
| + Ours | **0.457** | **0.449** | **2.077** | **0.914** | **0.306** | **0.347** | **0.193** | **0.296** | **0.628** | **0.345** | **0.288** | **0.314** |
| Promotion | 67.93% | | 57.30% | | 79.61% | | 30.32% | | 5.56% | | 56.16% | |
| Informer | 1.550 | 0.998 | 5.137 | 1.544 | 1.410 | 0.823 | 0.311 | 0.397 | 0.764 | 0.416 | 0.634 | 0.548 |
| + Ours | **0.496** | **0.460** | **2.125** | **0.928** | **0.460** | **0.434** | **0.226** | **0.330** | **0.719** | **0.409** | **0.275** | **0.302** |
| Promotion | 68.00% | | 58.63% | | 67.38% | | 27.33% | | 5.89% | | 56.78% | |
| Reformer | 1.280 | 0.932 | 4.724 | 1.443 | 1.479 | 0.915 | 0.338 | 0.429 | 0.741 | 0.423 | 0.803 | 0.656 |
| + Ours | **0.462** | **0.468** | **2.865** | **1.065** | **0.493** | **0.441** | **0.206** | **0.308** | **0.682** | **0.372** | **0.286** | **0.308** |
| Promotion | 63.91% | | 39.35% | | 66.67% | | 39.05% | | 7.96% | | 64.38% | |
| Autoformer | 0.613 | 0.539 | 3.006 | 1.161 | 0.324 | 0.368 | 0.227 | 0.338 | 0.628 | 0.379 | 0.338 | 0.382 |
| + Ours | **0.487** | **0.491** | **2.545** | **1.039** | **0.305** | **0.345** | **0.216** | **0.315** | **0.619** | **0.364** | **0.286** | **0.310** |
| Promotion | 20.55% | | 15.34% | | 5.86% | | 4.85% | | 1.43% | | 15.38% | |

## 4.2 Ablation Study

**Quality evaluation**  To explore the role of each module in our proposed framework, we compare the prediction results on ETTm2 obtained by three models: vanilla Transformer, Transformer with only Series Stationarization, and our Non-stationary Transformer. In Figure 3, we find out that the two modules strengthen the non-stationary forecasting ability of Transformer from different perspectives. Series Stationarization focuses on the alignment of statistical properties among each series input that benefits Transformer a lot to generalize on out-of-distribution data. However, as is shown in Figure 3(b), the over-stationarized circumstance for training makes the deep model more likely to output uneventful series with significant high stationarity and neglect the nature of non-stationary real-world data. With the aid of De-stationary Attention, the model gives concern back to the inherent non-stationarity of real-world time series. It is beneficial for an accurate prediction of the detailed series variation, which is vital in real-world time series forecasting.

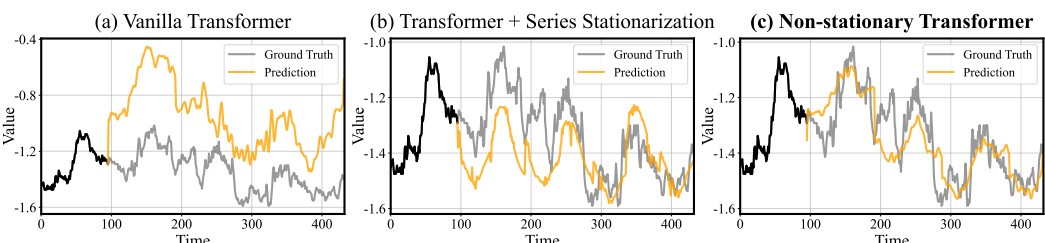

Figure 3: Visualization of ETTm2 predictions given by different models.

Table 5: Forecasting results obtained by applying different methods to Transformer and Reformer. We report the averaged MSE/MAE of all prediction lengths (stated in Table 2) for comparison. Complete results can be found in Appendix.

| Base Models | Transformer | | | | | | Reformer | | | | | |
|---|---|---|---|---|---|---|---|---|---|---|---|---|
| Methods | + RevIN [15] | | + Series Stationarization | | + **Ours** | | + RevIN [15] | | + Series Stationarization | | + **Ours** | |
| Metric | MSE | MAE | MSE | MAE | MSE | MAE | MSE | MAE | MSE | MAE | MSE | MAE |
| Exchange | 0.567 | 0.487 | 0.569 | 0.488 | **0.461** | **0.454** | 0.469 | 0.472 | 0.470 | 0.473 | **0.462** | **0.468** |
| ILI | 2.205 | 0.934 | 2.206 | 0.934 | **2.077** | **0.914** | 3.024 | 1.096 | 3.023 | 1.096 | **2.865** | **1.065** |
| ETTm2 | 0.460 | 0.416 | 0.461 | 0.416 | **0.306** | **0.347** | 0.542 | 0.459 | 0.537 | 0.459 | **0.493** | **0.441** |
| Electricity | 0.197 | 0.298 | 0.197 | 0.298 | **0.193** | **0.296** | 0.208 | 0.309 | 0.207 | 0.309 | **0.206** | **0.308** |
| Traffic | 0.643 | 0.352 | 0.641 | 0.352 | **0.628** | **0.345** | 0.687 | 0.378 | 0.691 | 0.380 | **0.682** | **0.372** |
| Weather | 0.301 | 0.316 | 0.304 | 0.317 | **0.288** | **0.314** | 0.291 | 0.309 | 0.292 | 0.309 | **0.286** | **0.308** |

**Quantitative performance**   In addition to the above case study, we also provide quantitative forecasting performance comparison with stationarization methods: a deep method RevIN [15] and Series Stationarization (Section 3.1). As is shown in Table 5, the forecasting results assisted by RevIN and Series Stationarization are basically the same, which indicates that the parameter-free version of normalization in our framework performs sufficiently to stationarize time series. Besides, the proposed De-stationary Attention in Non-stationary Transformers further boosts the performance and achieves the best in all six benchmarks. The MSE reduction brought by De-stationary Attention becomes significant, especially when the dataset is highly non-stationary (Exchange: $0.569 \rightarrow 0.461$, ETTm2: $0.461 \rightarrow 0.306$). The comparison reveals that simply stationarizing time series still limits the predictive capability of Transformers, and the complementary mechanisms in Non-stationary Transformers can properly release the models' potential for non-stationary series forecasting.

## 4.3   Model Analysis

**Over-stationarization problem**   To verify the over-stationarization problem from a statistical view, we train Transformers with the aforementioned methods respectively, arrange all predicted time series in chronological order and compare the degree of stationarity with the ground truth (Figure 4). While models solely equipped with stationarization methods tend to output series with unexpected high degree of stationarity, the results assisted by De-stationary Attention are close to the actual value (relative stationarity $\in [97\%, 103\%]$). Besides, as the degree of series stationarity increases, the over-stationarization problem becomes more significant. The huge discrepancy of the degree of stationarity can account for the inferior performance of Transformer with only stationarization. And it also demonstrates that De-stationary Attention as an internal renovation alleviates over-stationarization.

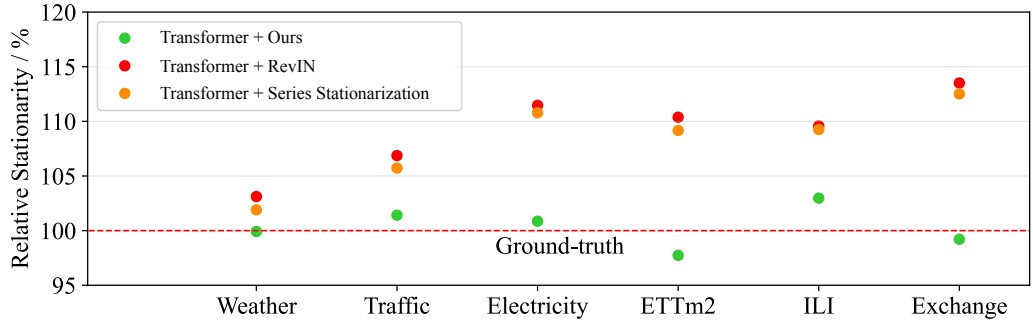

Figure 4: Relative stationarity is calculated as the ratio of ADF test statistics between the model predictions and ground truth. From left to right, the dataset is increasingly non-stationary. While models equipped with only stationarization tend to output highly stationary series, our method gives predictions with stationarity closer to ground truth.

**Exploring of Non-stationary Information Re-incorporation**  It is notable that by specifying over-stationarization as less distinguishable attention, we narrow down our design space into the attention calculation mechanism. To explore other approaches to retrieve non-stationary information, we conduct experiments by re-incorporating the $\mu$ and $\sigma$ into feed-forward layers (DeFF), which is the left part of the Transformer architecture. In detail, we feed learned $\mu$ and $\sigma$ into each feed-forward layer iteratively. As is shown in Table 6, re-incorporating non-stationarity is necessary only when the inputs are stationarized (Stationary), which is beneficial for forecasting but will lead to stationarity discrepancy of model outputs. And our proposed design (Stat + DeAttn) makes further promotion and achieves the best in most cases (77%). In addition to the theoretical analysis, experimental results further validate the effectiveness of our design in re-incorporating non-stationarity on attention.

Table 6: Ablation of framework design. *Baseline* means vanilla Transformer, *Stationary* means adding Series Stationarization, *DeFF* means re-incorporating non-stationarity on feed-forward layers, *DeAttn* means re-incorporating by De-stationary Attention, *Stat + DeFF* means adding Series Stationarization and re-incorporating on feed-forward layers. *Stat + DeAttn* means our proposed framework.

| Models | | Baseline | | Stationary | | DeFF | | DeAttn | | Stat + DeFF | | Stat + DeAttn | |
|---|---|---|---|---|---|---|---|---|---|---|---|---|---|
| Metric | | MSE | MAE | MSE | MAE | MSE | MAE | MSE | MAE | MSE | MAE | MSE | MAE |
| Exchange | 96 | 0.567 | 0.591 | 0.136 | 0.258 | 0.784 | 0.696 | 0.611 | 0.613 | 0.116 | 0.243 | **0.111** | **0.237** |
| | 192 | 1.150 | 0.825 | 0.239 | 0.348 | 1.162 | 0.866 | 1.202 | 0.840 | 0.280 | 0.383 | **0.219** | **0.335** |
| | 336 | 1.792 | 1.084 | 0.425 | 0.479 | 1.346 | 0.963 | 1.516 | 0.981 | **0.371** | **0.452** | 0.421 | 0.476 |
| | 720 | 2.191 | 1.159 | 1.475 | 0.865 | 2.042 | 1.163 | 2.894 | 1.377 | **0.934** | **0.704** | 1.092 | 0.769 |
| ILI | 24 | 4.748 | 1.430 | 2.573 | 0.980 | 4.850 | 1.445 | 4.734 | 1.424 | 2.404 | 0.985 | **2.294** | **0.945** |
| | 36 | 4.671 | 1.430 | 1.955 | 0.870 | 4.848 | 1.452 | 4.927 | 1.482 | 2.585 | 0.983 | **1.825** | **0.848** |
| | 48 | 4.994 | 1.482 | 2.057 | 0.902 | 4.903 | 1.466 | 4.996 | 1.483 | 2.496 | 0.991 | **2.010** | **0.900** |
| | 60 | 5.041 | 1.499 | 2.238 | 0.982 | 5.196 | 1.524 | 5.184 | 1.519 | 2.667 | 1.059 | **2.178** | **0.963** |
| ETTm2 | 96 | 0.572 | 0.552 | 0.253 | 0.311 | 0.767 | 0.635 | 0.304 | 0.406 | 0.275 | 0.329 | **0.192** | **0.274** |
| | 192 | 1.161 | 0.793 | 0.453 | 0.404 | 0.960 | 0.717 | 0.820 | 0.652 | 0.406 | 0.403 | **0.280** | **0.339** |
| | 336 | 1.209 | 0.842 | 0.546 | 0.461 | 1.159 | 0.811 | 1.406 | 0.883 | 0.502 | 0.465 | **0.334** | **0.361** |
| | 720 | 3.061 | 1.289 | 0.593 | 0.489 | 3.187 | 1.308 | 2.858 | 1.108 | 0.694 | 0.575 | **0.417** | **0.413** |
| Electricity | 96 | 0.260 | 0.358 | 0.171 | 0.275 | 0.260 | 0.356 | 0.253 | 0.351 | 0.170 | 0.274 | **0.169** | **0.273** |
| | 192 | 0.266 | 0.367 | 0.192 | 0.296 | 0.264 | 0.365 | 0.257 | 0.358 | 0.188 | 0.293 | **0.182** | **0.286** |
| | 336 | 0.280 | 0.375 | 0.208 | 0.306 | 0.277 | 0.374 | 0.270 | 0.365 | 0.206 | 0.309 | **0.200** | **0.304** |
| | 720 | 0.302 | 0.386 | **0.216** | **0.315** | 0.299 | 0.384 | 0.295 | 0.380 | 0.223 | 0.323 | 0.222 | 0.321 |
| Traffic | 96 | 0.647 | 0.357 | 0.614 | 0.337 | 0.646 | 0.353 | 0.650 | 0.358 | **0.605** | **0.333** | 0.612 | 0.338 |
| | 192 | 0.649 | 0.356 | 0.637 | 0.351 | 0.645 | 0.352 | 0.655 | 0.358 | 0.617 | 0.342 | **0.613** | **0.340** |
| | 336 | 0.667 | 0.364 | 0.653 | 0.359 | 0.672 | 0.360 | 0.656 | 0.355 | 0.635 | 0.349 | **0.618** | **0.328** |
| | 720 | 0.697 | 0.376 | 0.661 | 0.360 | 0.695 | 0.376 | 0.681 | 0.366 | **0.649** | **0.351** | 0.653 | 0.355 |
| Weather | 96 | 0.395 | 0.427 | 0.175 | 0.225 | 0.417 | 0.445 | 0.296 | 0.364 | 0.178 | 0.226 | **0.173** | **0.223** |
| | 192 | 0.619 | 0.560 | 0.273 | 0.297 | 0.699 | 0.604 | 0.480 | 0.464 | 0.256 | 0.295 | **0.245** | **0.285** |
| | 336 | 0.689 | 0.594 | 0.333 | **0.325** | 0.773 | 0.620 | 0.581 | 0.519 | 0.338 | 0.351 | **0.321** | 0.338 |
| | 720 | 0.926 | 0.710 | 0.436 | 0.420 | 1.008 | 0.718 | 0.795 | 0.642 | 0.417 | 0.412 | **0.414** | **0.410** |

## 5   Conclusion

This paper addresses time series forecasting from the view of stationarity. Unlike previous studies that simply attenuate non-stationarity leading to over-stationarization, we propose an efficient way to increase series stationarity and renovate the internal mechanism to re-incorporate non-stationary information, thus boosting data predictability and model predictive capability simultaneously. Experimentally, our method shows great generality and performance on six real-world benchmarks. And detailed derivations and ablations are provided to testify the effectiveness of each component in our proposed Non-stationary Transformers framework. In the future, we will explore a more model-agnostic solution for the over-stationarization problem.

## Acknowledgments

This work was supported by the National Key Research and Development Plan (2021YFC3000905), National Natural Science Foundation of China (62022050 and 62021002), Beijing Nova Program (Z201100006820041), and BNRist Innovation Fund (BNR2021RC01002).

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
