# OpenReview forum: "Non-stationary Transformers: Exploring the Stationarity in Time Series Forecasting"
_NeurIPS.cc/2022/Conference — NeurIPS 2022 Accept_

### Official Review · Reviewer_ZUaG · 2022-07-10

**Rating:** 7
**Confidence:** 4
**Soundness:** 3 good
**Presentation:** 3 good
**Contribution:** 4 excellent

**Summary:**

This paper focuses on forecasting and introduces a new method for scaling attention.
* The input raw series is chunk-normalized to yield the input of the backbone transformer model
* the chunk-wise scaling parameters are processed through MLPs to scale the rows of the attention matrix.

This procedure is showed to consistently improve performance of transformer-based forecasting models.

**Questions:**

* You should not use the words "stationary", "stationarity", "de-stationary" etc unless this is actually what you mean.
* From a broad perspective, I see your proposed method as applying some chunk-wise transform, say x', \theta = F(x), that returns some modified version for a chunk along with some parameters, and then using some MLP1(\theta) and MLP2(\theta) for scaling the attention matrix. At the output, you apply F^{-1}(out, \theta). I genuinely think this is a great idea, and I think it is ok to motivate this by the following story:
   - Having all chunks have the same first two moments (mean and variance) is a first step in enforcing them to have the same distribution and this is what we do with our series normalization. In this case, our chunkwise transform is a simple normalization, and \theta={\mu, \sigma}.
   - Imagining that we do applied just some linear transform on the input, here's what the attention matrix would be like. So we propose some scaling scheme for the attention inspired by this, that looks like.....
- at the output, we can transform back.
   The advantage of this  way of writing things is that you leave room for further research (notably more sophisticated chunkwise methods) and you don't wrongly pretend that normalizing means enforcing stationarity... because it's not the case.

I think that this paper really brings great things on the table, but that it should undergo some strong modifications to be ok. The good news in my view is that I am not suggesting any change to the actual model, which is good, but just to the story that is told regarding the theory, so that I believe this is feasible (although it requires some heavy work).


Below, please find a list of comments on the go.
Title
* I personally don't like all these "rethinking *" titles, that sound very pretentious in my view, somehow suggesting that things were not correctly thought about before and that everything will change from now on. Please note that your view of stationarity is not rigorous, adding a bit to my frustration.

Introduction:
* "From the aspect of data": awkward. Furthermore, this statement about predictability is not very well motivated in my opinion.
* "From the view of deep learning": why would that be limited to deep learning ?
* "a hot topic": this is an exagerated statement, provided you give 3 references, among which one is almost 15 years old.
* Figure 1 is not clear enough in my opinion. I don't really understand what the input data to the transformer is: are these the I, II, III chunks ? What is that the second column ? Just zooms for the data ? And finally, what are these "attention matrices" ? what's the number of rows, columns. I suspect some interpolation for these images ? Are you using some `interpolation='nearest'` for your plot ?

Related work
* "which is the essential property": an essential
* "is the key to predictability": a key. I am not sure this statement is supported by such generic references.
* I am a bit annoyed by the fact that you constantly talk of stationarity and non-stationarity, but you don't define the terms.


Non-stationary Transformers
* "the key to time series predictability": again, too strong. "an important element" ?
* The reference to figure 1 is not clear. As mentioned already I don't really see what Figure 1 brings on the table exactly, maybe just because it is poorly explained and I don't understand it.
* "To deal with the dilemna [...] simultaneously": you wrote that already
* although this is classical to use "T" for transposition, please note that using \top instead gives a better rendering.

* I really must protest against your statement that a simple normalization is enough to "transform it into standard Gaussian distribution". I don't see any connection between having x be centered and unit variance and x being N(0,1). Of course, the first two moments are matching, but this is not at all sufficient to make x Gaussian. You should remove this statement and simply mention that you are centering and normalizing it.
* I understand the use of "Hadamard product", but the fact is that it can be confusing to some readers. If you like, you could be using a statement like "\frac{a}{b} and a\circ b are element-wise division and product, respectively".
* I don't understand how these \mu_x and \sigma_x vectors are computed. From a single S-dimensional chunk, how do you end up computing a S-dimensional mean and a S-dimensional variance statistic ? Checking out the followup, it turns out you are simply normalizing each chunk, so that \mu_x and \sigma_x are just scalars. You should not be introducing this useless notation.
* Note that Normalization module eliminates the [...] statistics": I don't see at all how this statement is supported. The only thing you enforce is that every chunk is normalized. You may not at all write that this leads to all of them having the same "statistics". As you know, you can't seriously write that mean and standard deviation are the only statistics for a time series... What about all the auto-correlation structure, higher-order statistics, etc. I am fine with your normalization, but please don't write that it results in chunks having the same distribution ! You are simply matching the first two moments.

* You didn't explain how the "sliding window" is achieved: is there some overlap between your chunks ? Or do you obtain your chunks with a simple folding operation (window size=hop size)
* I don't see the point in using uppercase "Series Stationarization" if you're not going to use the (unfortunate) acronym SS. Maybe just use lower case then.
* "the base models will receive stationarized inputs": again, I feel uncomfortable with accepting the statement that a series of normalized chunks is a stationary time series. Just imagining a dataset with outliers, your procedure will obviously change the content of a chunk containing the outlier in a undesired way, so that future research might focus on some other transformation than mean-variance normalization, like quantiles or such. Don't misread me: the method is nice, but in my view you are claiming something that is not exactly there.
* "for an instance": for instance
* again, the reference to figure 1 is unclear to me, because I didn't get what you want exactly to be conveying with this figure
* "following the same distribution": it is not because they are both centered and unit variance that two vectors have the same distribution !
* "to tackle the trivial attention": awkward. to tackle this over-stationarization problem ?
* "To simplify the analysis [...] input series": this assumption is completely unreasonable and it is useless in my view to rely on it seriously for anything. This said, I understand that assuming f to be linear allows you to derive the *general form* of your "de-stationary attention", as what would be happening in the linear case. The fact is that you are actually not using this assumption in the followup since your attention module don't use these \mu and \sigma as such, but instead through some MLPs. I think that the story would be more readable if you introduced it as you do in the linear case, but then simply propose this attention scaling procedure ase a way to reinject the normalization parameters back into attention in a way that generalizes the linear case. This would read much better to me.
* "Since it is a convention to conduct [...] is reduced to a scalar". This sentence makes no sense to me. Just take \mu_x and \sigma_x as scalars right from the start and be good with this.
* As stated above, it is not necessary to write that your \tau and \Delta are there to "approximate \sigma_x^2 and K\mu_q" (you know them already !) but rather that you introduce them as scaling factors in analogy to what happens in the linear case. Don't forget that your derivations assume a linearity that is not there.


Experiments
* What do you call "stationarity" and "relative stationarity" for the "over-stationarization problem" section ? This is not defined.


Conclusion
* "The impressive generality and performance of the proposed framework": please change this pretentious statement.


**Limitations:**

The authors are a bit pretentious at times, but other than that it is all right

**Strengths And Weaknesses:**

Strengths
* The proposed procedure is quite simple to implement
* In my view, introducing this attention scaling is actually a nice and elegant thing to do.
* The proposed scaled-attention mechanism obviously helps a lot in practice to improve performance

Weaknesses
* overall, I would say that the theoretical part is not very rigorous.
* The paper focuses its whole discussion on the concept of "stationarity", which is a pretty well defined word with a strong historical background. basically, a time series x is stationary if time shifts do not change its distribution, but the use that is made here is. Here, the authors call a time series stationary if all (vector) samples are normalized. This is not related.

-----
EDIT after reviews and discussions

I believe the authors did an amazing job during this review round and I actually think this paper has some big potential impact, even if I still have many things to say regarding its rigour, but I think this is not a fundamental flow that should prevent the paper from being published.

---

> ### Author Response · Authors · 2022-08-01
> **Response to Reviewer ZUaG (Part 1)**
>
> We would like to sincerely thank Reviewer ZUaG for providing a detailed review and insightful suggestions.
>
> **Q1:** Rephrase and specific the concepts of "stationary", "stationarity", "relative stationarity" and "de-stationary".
>
> Thanks a lot for your suggestion with scientific rigor. We have updated all the usages of the above concepts in the $\underline{\text{revised paper}}$. In detail, we reclarify and define the following concepts.
>
> - The degree of stationarity: a value to measure the degree of distribution change in time series. Especially, in this paper, we adopt the ADF test statistic as the metric. A smaller ADF test statistic indicates a higher degree of stationarity, which means the distribution is more stable.
> - Stationarization: A method to increase the degree of stationarity.
> - De-stationarization/De-stationary: A method to decrease the degree of stationarity back for stationarized time series.
> - Relative stationarity in Figure 4: The ratio of the ADF test statistic between the ground truth time series and the model predictions.
>
> Especially, for the stationarization, we do not attempt to make the raw time series completely stationary. What we try is to increase the degree of stationarity, that is making the time series distribution more stable.
>
> Concretely, for the normalization module, we agree that the normalization based on $\mu$ and $\sigma$ cannot make the raw data a stationary time series. But with our normalization module, the ADF test statistic of the time series is getting smaller, which means the time series distribution is more stable and the time series "tends more to be stationary". This verifies that our proposed normalization module is an effective design to increase the degree of stationarity.
>
> |ADF Test Statistic|Exchange|ILI|ETTm2|Electricity|Traffic|Weather|
> |-|-|-|-|-|-|-|
> |Raw data|-1.889|-5.406|-6.225|-8.483|-15.046|-26.661|
> |After our Normalization|-9.937|-10.313|-33.485|-20.888|-18.946|-35.010|
>
> **Q2:** Rephrase the theoretical part.
>
> Following the reviewer's suggestion, we have completely polished the theoretical part in the $\underline{\text{revised paper}}$.
>
> (1) Revised parts.
>
> - Formulations and descriptions for equations, especially the Hadamard product and transpose.
> - Reconsider and deliberate all the assumptions in our plain model analysis.
>
> (2) Unchanged parts, especially the linear property of $f$.
>
> As mentioned by the reviewer, the linear property of $f$ will derive the form of De-stationary Attention in $\underline{\text{Equation 5 of main text}}$. We would like to emphasize that this formalization is highly instructive for our final design.
>
> - Note that there are many ways to reinject the normalization parameters back into attention, such as only $\tau$, only $\mathbf{\Delta}$ or other combination ways. If we didn't have the formalization in $\underline{\text{Equation 5 of main text}}$, we would have to try plenty of designs. Thus, the linear property of $f$ motivates us with tractable design space.
>
> - Besides, we also provide an ablation study in $\underline{\text{Table 9 of supplementary materials}}$, which compares the only $\tau$ and only $\mathbf{\Delta}$ designs. The results confirm that the formalization in $\underline{\text{Equation 5 of main text}}$ achieves the best performance, demonstrating that the derived formalization surpasses other designs without such a nice theoretical support.
>
> Thus, in view of the instruction of linear property for our final design, we still hold the linear property assumption, which can also make the reader easier to understand our design in De-stationary Attention.
>
> Besides, we would like to keep the original descriptions in "Approximate $\sigma_{\mathbf{x}}^2$ and $\mathbf{K}\mu_\mathbf{Q}$". Note that even in the linear model analysis, both $\mathbf{K}$ and $\mu_\mathbf{Q}$ cannot be obtained directly without the parameter weights in $f$. Thus, we adopt the MLPs to approximate them and obtain $\tau$ and $\mathbf{\Delta}$.

---

> > ### Author Response · Authors · 2022-08-01
> > **Response to Reviewer ZUaG (Part 2)**
> >
> > **Q3:** The "sliding window" operation and calculation in Series Stationarization.
> >
> > The "sliding window" design is a convention of the time series forecasting applications. For example, in deployment, the model input is $S$ time points. As time goes by, we will keep sliding the input segment to obtain the latest length-$S$ past series for forecasting. We use this description to illustrate that our normalization is conducted on an input segment, not the whole time series.
> >
> > The detailed calculation of Series Stationarization is in the $\underline{\text{Algorithm 1 and Algorithm 2 of supplementary materials}}$. For the length-$S$ time series segment $\mathbf{x}\in\mathbb{R}^{S\times C}$ with $C$ dimension, the normalization is conducted to each dimension. Thus, both $\mu_\mathbf{x}$ and $\sigma_\mathbf{x}$ are vectors.
> >
> > Since both $\mu_\mathbf{x}$ and $\sigma_\mathbf{x}$ are vectors, we assume that each variable of time series $\mathbf{x}$ shares the same variance for the convenience of derivation. This assumption is reasonable because the z-score normalization is widely used as the preprocessing of time series forecasting ($\underline{\text{lines 160-162 of main text}}$).
> >
> > **Q4:** Suggestions on storytelling.
> >
> > We very much appreciate your great suggestions about the storytelling, which can make the technique much clear and easier to understand. While we do improve the other parts, we would like to maintain the motivation about the "stationarization" because of the following questions.
> >
> > - Our method is indeed motivated by the insights and analyses of the "stationarity" of time series. The over-stationarization problem is specified and supported by the ADF test statistic in $\underline{\text{Figure 4 of main text}}$. The ADF test statistic reflects that the previous model predictions do have larger degrees of stationarity than the ground truth. All these analyses guide us to think about the "stationarity" of time series.
> > - To resolve the reviewer's concern about the misleading usage of the "stationarity", we have rephrased all the corresponding usages based on the "degree of stationarity", which is clearly defined in $\underline{\text{Q1}}$. The $\underline{\text{revised paper}}$ does not have misleading usages about the "stationarity" and "non-stationarity".
> > - As you stated, we attempt to "bring great things on the table". And stationarity is an important element in time series analysis. We hope that our writing about "stationarity" can provide some guidance for future research on time series forecasting. Please let us know if the above improvement is still questionable in your view.
> >
> >
> >
> > **Q5:** Pretentious statement in title and conclusion.
> >
> > We have updated the title as "Non-stationary Transformers: Exploring the Stationarity in Time Series Forecasting" and removed the "impressive generality and performance of the proposed framework" in the conclusion in the $\underline{\text{revised paper}}$.
> >
> >
> >
> > **Q6:** Writing issues.
> >
> > All the following writing issues have been rephrased in the $\underline{\text{revised paper}}$.
> >
> > - We have polished the literature review about the time series stationarity in the Introduction.
> > - The grammatical mistakes and semantic repetitions are resolved.
> > - We have rephrased the less rigorous and wrong expressions. For example, normalization does not transform the distribution into Gaussian distribution; normalization cannot eliminate the statistics difference; matching the first two moments cannot make the time series follow the same distribution.
> > - Since the Series Stationarization is the set of normalization and de-normalization, we use the upper case for Series Stationarization to represent the combination of these two modules.
> >
> >
> >
> > **Q7:** The descriptions about Figure 1.
> >
> > As stated by the reviewer, the input data to the Transformer is the I, II, III chunks respectively. The second column is just zoom-ins for the data. For clearness, we have colored the background and added the text descriptions to the figure in the $\underline{\text{revised paper}}$.
> >
> > As for the attention matrices, it is the visualization of the calculated attention maps. For a length-$S$ time series, the matrices are in the shape of $S\times S$. Thus, the value in the $i$-th row and $j$-th column represents the normalized attention weight of $i$-th time point w.r.t. the $j$-th time point.
> >
> > We would like to thank the reviewer's meticulous suggestions again. All the suggestions are included in our $\underline{\text{revised paper}}$.

---

### Official Review · Reviewer_8mJi · 2022-07-11

**Rating:** 7
**Confidence:** 4
**Soundness:** 3 good
**Presentation:** 4 excellent
**Contribution:** 4 excellent

**Summary:**

This paper proposes Non-stationary Transformers as a generic framework to tackle over-stationarization problem. It includes Series Stationarization and De-stationary Attention module, where Series Stationarization converts raw time series into more stationary ones for better predictability and De-stationary Attention is devised to recover the intrinsic non-stationary information in raw time series during self-station stage. Results show that Non-stationary Transformers are generally applicable with various Transformers and significantly improve results compared to the existing state-of-the-art.

**Questions:**

What is the standard deviation of results listed in the paper? It will be good to know how statistically significant the proposed method is.

**Limitations:**

Figure 4 requires readers to re-read Table 1 "ADF Test Statistics" column. Authors can consider adding information in figure 4 to clarify that datasets are sorted according to ADF Test Statistics so that readers are easier to follow.

**Strengths And Weaknesses:**

**Strengths**
* The motivation is clear and the writing is easy to follow. The De-stationary Attention module is quite simple and clearly deduced in Equation 5, making its implementation straightforward and easy to understand.
* The results are quite impressive and simple techniques can significantly improve over various state-of-the-art baselines.

**Weaknesses**
* The proposed Series Stationarization technique share lots of similarity with section 3.3 Scale handling in [1]. The authors need clearly discuss their difference.
* $\tau$ and $\triangle$ are learned in the paper. However, if  $\tau$ and $\triangle$ are used to approximate the variance and mean term, authors need to show if directly utilizing pre-computed statistics will work. Such an ablation study will make this work more solid.

Reference
[1] David Salinas, Valentin Flunkert, Jan Gasthaus. DeepAR: Probabilistic Forecasting with Autoregressive Recurrent Networks.

---

> ### Author Response · Authors · 2022-08-01
> **Response to Reviewer 8mJi (Part 1)**
>
> Many thanks to Reviewer 8mJi for providing a detailed review and insightful questions.
>
> **Q1:** The difference between Series Stationarization technique and the Scale handling in DeepAR.
>
> For a segment of time series $\mathbf{x}\in\mathbb{R}^{T\times C}$ with mean $\mu_{\mathbf{x}}\in\mathbb{R}^{1\times C}$ and variance $\sigma_{\mathbf{x}}\in\mathbb{R}^{1\times C}$, the normalization in Series Stationarization is $\frac{\mathbf{x}-\mu_{\mathbf{x}}}{\sigma_{\mathbf{x}}}$ and the normalization in Scale handling is $\frac{\mathbf{x}}{\mu_{\mathbf{x}}+1}$.
>
> We demonstrate that the differences lie in the following two folds:
>
> (1) The differences in motivation and effectiveness.
>
> Our proposed Series Stationarization focuses on the time series stationarity, while the Scale handling attempts to balance the value ranges of multiple time series. The detailed comparison of the module effectiveness is shown as follows:
>
> - **Both designs can balance the value ranges of multiple time series.** Especially, Series Stationarization and Scale handling transform the mean of multiple time series into zero and almost one respectively.
>
> - **Our proposed Series Stationarization is more powerful in enhancing the time series stationarity,** thereby matching the problem that our paper addresses. The comparison of ADF test statistic is shown as follows. Note that a smaller value of ADF Test Statistic means more likely to be stationarity.
>
> |ADF test statistic|Exchange|ILI|ETTm2|Electricity|Traffic|Weather|
> |-|-|-|-|-|-|-|
> |Raw data|-1.889|-5.406|-6.225|-8.483|-15.046|-26.661|
> |After Scale handling|-1.915|-5.527|-6.435|-8.560|-15.025|-27.058|
> |After Series Stationarization|**-9.937**|**-10.313**|**-33.485**|**-20.888**|**-18.946**|**-35.010**|
>
> In addition, as stated in the $\underline{\text{lines 135-137 of main text}}$, Series Stationarization can make the model equivariant to translational and scaling perturbance of time series, thereby benefiting non-stationary series forecasting. In contrast, the Scale handling does not maintain this equivariance.
>
> (2) Experimental comparison.
>
> To further compare the effect of these two modules in predictive capability, we replace the Series Stationarization in Non-stationary Transformer as the Scale handling. Benefiting from the stronger capability in enhancing the time series stationarity, Series Stationarization generally outperforms the Scale handling.
>
> |Exchange (MSE\|MAE)|Predict 96|Predict 192|Predict 336|Predict 720|
> |-|-|-|-|-|
> |Vanilla Transformer|0.567 \| 0.591 | 1.150 \| 0.825 |1.792 \| 1.084 | 2.191 \| 1.159 |
> |+ Scale handling|0.237 \| 0.380|0.516 \| 0.576|0.737 \| 0.706|1.413 \| 1.009|
> |+ Series Stationarization (Ours)|**0.111** \| **0.237**|**0.219** \| **0.335**|**0.421** \| **0.476**|**1.092** \| **0.769**|
>
> |ILI (MSE\|MAE)|Predict 24|Predict 36|Predict 48|Predict 60|
> |-|-|-|-|-|
> |Vanilla Transformer|4.748 \| 1.430|4.671 \| 1.430|4.994 \| 1.482|5.041 \| 1.499|
> |+ Scale handling|3.276 \| 1.125|3.629 \| 1.192|3.730 \| 1.245|3.661 \| 1.238|
> |+ Series Stationarization (Ours)|**2.294** \| **0.945**|**1.825** \| **0.848**|**2.010**\| **0.900**| **2.178** \| **0.963**|
>
> |ETTm2 (MSE\|MAE)|Predict 96|Predict 192|Predict 336|Predict 720|
> |-|-|-|-|-|
> |Vanilla Transformer|0.572 \| 0.552| 1.161 \| 0.793|1.209 \| 0.842| 3.061 \| 1.289|
> |+ Scale handling|0.379 \| 0.462|0.919 \| 0.762|1.875 \| 1.140|3.832 \| 1.509|
> |+ Series Stationarization (Ours)|**0.192** \| **0.274**| **0.280** \| **0.339**|**0.334** \| **0.361**| **0.417** \| **0.413**|
>
> |Electricity (MSE\|MAE)|Predict 96|Predict 192|Predict 336|Predict 720|
> |-|-|-|-|-|
> |Vanilla Transformer|0.260 \| 0.358| 0.266 \| 0.367| 0.280 \| 0.375|0.302 \| 0.386|
> |+ Scale handling|0.282 \| 0.378|0.323 \| 0.411|0.340 \| 0.423|0.336 \| 0.417|
> |+ Series Stationarization (Ours)|**0.169** \| **0.273**| **0.182**\| **0.286**| **0.200** \| **0.304**| **0.222** \| **0.321**|
>
> |Traffic (MSE\|MAE)|Predict 96|Predict 192|Predict 336|Predict 720|
> |-|-|-|-|-|
> |Vanilla Transformer|0.647 \| 0.357|0.649 \| 0.356|0.667 \| 0.364| 0.697 \| 0.376|
> |+ Scale handling|0.798 \| 0.508|0.740 \| 0.417|0.760 \| 0.448|0.878 \| 0.476|
> |+ Series Stationarization (Ours)|**0.612** \| **0.338** | **0.613** \| **0.340**| **0.618** \| **0.328** | **0.653** \| **0.355**|
>
> |Weather (MSE\|MAE)|Predict 96|Predict 192|Predict 336|Predict 720|
> |-|-|-|-|-|
> |Vanilla Transformer| 0.395 \| 0.427 | 0.619 \| 0.560 | 0.689 \| 0.594 | 0.926 \| 0.710 |
> |+ Scale handling|0.248 \| 0.339|0.334 \| 0.412|1.157 \| 0.800|0.969 \| 0.732|
> |+ Series Stationarization (Ours)|**0.173** \| **0.223**|**0.245** \| **0.285**|**0.321** \| **0.338**|**0.414** \| **0.410**|

---

> > ### Author Response · Authors · 2022-08-01
> > **Response to Reviewer 8mJi (Part 2)**
> >
> > **Q2:** The ablation study of utilizing pre-computed statistics.
> >
> > Based on the derivation in $\underline{\text{Equation 5 of main text}}$, the optimal values of $\tau$ and $\mathbf{\Delta}$ are $\sigma_{\mathbf{x}}^2$ and $\mathbf{K}\mu_{\mathbf{Q}}$ respectively. These parameters are **data-dependent and rely on the deep features.** Thus, we cannot pre-compute the ground truth statistics of $\tau$ and $\mathbf{\Delta}$, that is why we adopt one MLP with current inputs to approximate them.
> >
> > To further address the reviewer's concern, we create a new well-designed baseline as follows:
> >
> > We still use the Non-stationary Transformer but add a well-trained parallel Transformer, where the former is with Series Stationarization and the latter inputs the non-stationarized raw data. For this new baseline, we calculate the "optimal values" of $\tau$ and $\mathbf{\Delta}$ from the well-trained parallel Transformer and use the calculated $\tau$ and $\mathbf{\Delta}$ to refine the attention. From the below results, we find that our original design surpasses the new baseline with a parallel Transformer, even though the latter is twice larger in parameter and computation cost.
> >
> > |Exchange (MSE\|MAE)|Predict 96|Predict 192|Predict 336|Predict 720|
> > |-|-|-|-|-|
> > |Vanilla Transformer|0.567 \| 0.591 | 1.150 \| 0.825 |1.792 \| 1.084 | 2.191 \| 1.159 |
> > |+ Parallel Transformer|0.162 \| 0.280| 0.228 \| 0.350|**0.373** \| **0.449**| 1.579 \| 0.898|
> > |+ Ours|**0.111** \| **0.237**|**0.219** \| **0.335**|0.421 \| 0.476|**1.092** \| **0.769**|
> >
> > |ILI (MSE\|MAE)|Predict 24|Predict 36|Predict 48|Predict 60|
> > |-|-|-|-|-|
> > |Vanilla Transformer|4.748 \| 1.430|4.671 \| 1.430|4.994 \| 1.482|5.041 \| 1.499|
> > |+ Parallel Transformer|3.426 \| 1.193|3.826 \| 1.247| 3.886 \| 1.281| 3.324 \| 1.195|
> > |+ Ours|**2.294** \| **0.945**|**1.825** \| **0.848**|**2.010** \| **0.900**| **2.178** \| **0.963**|
> >
> > |ETTm2 (MSE\|MAE)|Predict 96|Predict 192|Predict 336|Predict 720|
> > |-|-|-|-|-|
> > |Vanilla Transformer|0.572 \| 0.552| 1.161 \| 0.793|1.209 \| 0.842| 3.061 \| 1.289|
> > |+ Parallel Transformer|0.230 \| 0.302| 0.336 \| 0.357| 0.459 \| 0.424| 0.547 \| 0.475|
> > |+ Ours|**0.192** \| **0.274**| **0.280** \| **0.339**|**0.334** \| **0.361**| **0.417** \| **0.413**|
> >
> > |Electricity (MSE\|MAE)|Predict 96|Predict 192|Predict 336|Predict 720|
> > |-|-|-|-|-|
> > |Vanilla Transformer|0.260 \| 0.358| 0.266 \| 0.367| 0.280 \| 0.375|0.302 \| 0.386|
> > |+ Parallel Transformer|0.170 \| 0.275 | 0.196 \| 0.299| 0.226 \| 0.320| 0.227 \| 0.322|
> > |+ Ours|**0.169** \| **0.273**| **0.182** \| **0.286**| **0.200** \| **0.304**| **0.222** \| **0.321**|
> >
> > |Traffic (MSE\|MAE)|Predict 96|Predict 192|Predict 336|Predict 720|
> > |-|-|-|-|-|
> > |Vanilla Transformer|0.647 \| 0.357|0.649 \| 0.356|0.667 \| 0.364| 0.697 \| 0.376|
> > |+ Parallel Transformer|0.613 \| 0.334| 0.627 \| 0.343| 0.623 \| 0.337| 0.663 \| 0.357|
> > |+ Ours|**0.612** \| **0.338** | **0.613** \| **0.340**| **0.618** \| **0.328** | **0.653** \| **0.355**|
> >
> > |Weather (MSE\|MAE)|Predict 96|Predict 192|Predict 336|Predict 720|
> > |-|-|-|-|-|
> > |Vanilla Transformer|0.395 \| 0.427 | 0.619 \| 0.560 | 0.689 \| 0.594 | 0.926 \| 0.710 |
> > |+ Parallel Transformer|0.213 \| 0.261|0.265 \| 0.307| 0.332 \| 0.351|0.478 \| 0.436|
> > |+ Ours| **0.173** \| **0.223**|**0.245** \| **0.285**|**0.321**\| **0.338**|**0.414** \| **0.410**|
> >
> > **Q3:** The standard deviation of the results.
> >
> > We repeat each experiment three times with different random seeds. The standard deviations for the main results are provided in the $\underline{\text{Table 4 of supplementary materials}}$.
> >
> >
> > **Q4:** Description of Figure 4.
> >
> > Thanks for this valuable suggestion. We have added the information about the ADF test statistic in the $\underline{\text{revised paper}}$.

---

### Official Review · Reviewer_q3Tk · 2022-07-12

**Rating:** 4
**Confidence:** 4
**Soundness:** 2 fair
**Presentation:** 3 good
**Contribution:** 2 fair

**Summary:**

This paper study the non-stationary problem for transformer based forecasting model. They find that the nomarlization step in pre-process will lead transformers to generate indistinguishable temporal attention, which harms the prediction capability. To address this issue, the authors propose the de-stationary attention mechanism which considers the mean and variance statistics when computing the attention score. Empirical studies show consistent improvements over different type of transformers.

**Questions:**

see above

**Limitations:**

see above

**Strengths And Weaknesses:**

Pros:
1. The paper is easy to read and the structure is well organized.
2. The proposed strategy can be easily integrate with different transformer backbone.
3. Empirical results show consistent improvement.

Cons:
1. The proposed method is not very convincing. I appreciate the author for presenting the analysis for vanilla self-attention in section 3.2. However, I am confused for the motivation of using normalization layer. If the normalized data lead to the vanishment of so-called non-stationary information, why not remove the normalization before embedding layer and use the raw data for computation of attention score?
2. My major concern is the experiment setting is unfair. The standard preprocess protocol for long sequence forecasting includes the zero-mean normalized, but the reported RMSE, MAE has not inverted the forecasting results. It is unclear the coupled effect for the proposed normalization strategy and the normalization in preprocessing steps. It would be more convincing if the authors could compare the prediction on the original space with the scale-based evaluation metric, like sMAPE, MASE, etc.
3. It would be better to incorporate more advanced transformed based forecasting model, like ETSformer, FEDformer ( since these models has beaten autoformer with a margin), and justify the new normalization can also bring performance boost.
4. There exist some open question for the experiments on long sequence forecasting setting. It would be more convincing if the author can present the hyper-parameter selection strategy for the baselines to guarantee the fair comparison, for example, the hyper-parameters for N-Beats and LSTNet.

---

> ### Author Response · Authors · 2022-08-01
> **Response to Reviewer q3Tk (Part 1)**
>
> Many thanks to Reviewer q3Tk for providing the thorough insightful comments.
>
> **Q1:** The necessity of Series Stationarization.
>
> (1) Literature analysis.
>
> As we stated in the $\underline{\text{lines 29-34 of the main text}}$, the non-stationary time series can affect the prediction in both data and deep learning views.
>
> - "The non-stationary time series is less predictable", which is a basic idea of time series analysis.
> - "Non-stationary time series corresponds to the change of statistical properties and joint distributions over time. It is a fundamental but challenging problem to make deep models generalize well on a varying distribution."
>
> Thus, **Series Stationarization is well-supported by the common knowledge of time series analysis and deep learning**. Further, since the direct stationarization will lead to the over-stationary problem, we design the Non-stationary Transformer with the independent modules: Series Stationarization and De-stationary Attention. This design can make the model receive the stationarized inputs and avoid the over-stationary outputs simultaneously.
>
> (2) Experimental analysis.
>
> We have provided an ablation study in $\underline{\text{Table 5 of main text}}$ and $\underline{\text{Table 6 of supplementary materials}}$, where we can find that the Series Stationarization can benefit the time series forecasting. Especially, with Series Stationarization, the input-96-predict-336 MSE of Reformer changes from 1.549 to 0.613 on ETTm2 and from 1.357 to 0.426 on Exchange.
>
> To further address your concern, we newly add an ablation study using the raw data for the computation of attention score. It means that we remove the Series Stationarization and only maintain the De-stationary Attention. Here are the results. We can find that Series-stationarization can bring benefits to all benchmarks. For the datasets with stronger non-stationarity, the benefit of Series-stationarization becomes more significant.
>
> |Exchange (MSE\|MAE)|Predict 96|Predict 192|Predict 336|Predict 720|
> |-|-|-|-|-|
> |Vanilla Transformer|0.567 \| 0.591 | 1.150 \| 0.825 |1.792 \| 1.084 | 2.191 \| 1.159 |
> | + only De-stationary Attention| 0.611 \| 0.613|1.202 \| 0.840|1.516 \| 0.981 | 2.894 \| 1.377|
> | + Ours|**0.111** \| **0.237**|**0.219** \| **0.335**|**0.421** \| **0.476**|**1.092** \| **0.769**|
>
> |ILI (MSE\|MAE)|Predict 24|Predict 36|Predict 48|Predict 60|
> |-|-|-|-|-|
> |Vanilla Transformer|4.748 \| 1.430|4.671 \| 1.430|4.994 \| 1.482|5.041 \| 1.499|
> | + only De-stationary Attention|4.734 \| 1.424|4.927 \| 1.482| 4.996 \| 1.483|5.184 \| 1.519|
> | + Ours|**2.294** \| **0.945**|**1.825** \| **0.848**|**2.010**\| **0.900**| **2.178** \| **0.963**|
>
> |ETTm2 (MSE\|MAE)|Predict 96|Predict 192|Predict 336|Predict 720|
> |-|-|-|-|-|
> |Vanilla Transformer|0.572 \| 0.552| 1.161 \| 0.793|1.209 \| 0.842| 3.061 \| 1.289|
> | + only De-stationary Attention|0.304 \| 0.406| 0.820 \| 0.652| 1.406 \| 0.883| 2.858 \| 1.108|
> | + Ours|**0.192** \| **0.274**| **0.280** \| **0.339**|**0.334** \| **0.361**| **0.417** \| **0.413**|
>
> |Electricity (MSE\|MAE)|Predict 96|Predict 192|Predict 336|Predict 720|
> |-|-|-|-|-|
> |Vanilla Transformer|0.260 \| 0.358| 0.266 \| 0.367| 0.280 \| 0.375|0.302 \| 0.386|
> | + only De-stationary Attention|0.253 \| 0.351|0.257 \| 0.358| 0.270 \| 0.365 |0.295 \| 0.380|
> | + Ours|**0.169** \| **0.273**| **0.182**\| **0.286**| **0.200** \| **0.304**| **0.222** \| **0.321**|
>
> |Traffic (MSE\|MAE)|Predict 96|Predict 192|Predict 336|Predict 720|
> |-|-|-|-|-|
> |Vanilla Transformer|0.647 \| 0.357|0.649 \| 0.356|0.667 \| 0.364| 0.697 \| 0.376|
> | + only De-stationary Attention|0.650 \| 0.358| 0.655 \| 0.358| 0.656 \| 0.355|0.681 \| 0.366|
> | + Ours|**0.612** \| **0.338** | **0.613** \| **0.340**| **0.618** \| **0.328** | **0.653** \| **0.355**|
>
> |Weather (MSE\|MAE)|Predict 96|Predict 192|Predict 336|Predict 720|
> |-|-|-|-|-|
> |Vanilla Transformer| 0.395 \| 0.427 | 0.619 \| 0.560 | 0.689 \| 0.594 | 0.926 \| 0.710 |
> | + only De-stationary Attention| 0.296 \| 0.364| 0.480 \| 0.464 | 0.581 \| 0.519 | 0.795 \| 0.642|
> | + Ours|**0.173** \| **0.223**|**0.245** \| **0.285**|**0.321** \| **0.338**|**0.414** \| **0.410**|

---

> > ### Author Response · Authors · 2022-08-01
> > **Response to Reviewer q3Tk (Part 2)**
> >
> > **Q2:** The concern of the experiment setting.
> >
> > (1) All the methods in our paper are compared under consistent and fair evaluation protocol.
> >
> > As the reviewer stated, it is a convention in long sequence forecasting, that is including the zero-score normalization and reporting the MSE and MAE on the zero-score normalized time series.
> >
> > Note that in the experiments of the Non-stationary Transformer, **in pursuit of a fair comparison with previous methods, the zero-score normalization is also adopted as the preprocessing.** Thus, although Non-stationary Transformer adopts the Series Stationarization, both the prediction and ground truth for evaluation are still in the zero-score normalized time series space, which is the same as the convention setting.
> >
> > Thus, in our paper, all the methods are evaluated based on the same benchmark and protocol, thereby presenting a fair comparison.
> >
> > (2) Compare the prediction on the original space with the scale-based evaluation metrics.
> >
> > To further address the reviewer's concern about the "coupled effect for the proposed normalization strategy and the normalization in preprocessing steps", we provide new comprehensive results on the original space. For sMAPE (in the range of $[0,200]$) and MASE, we follow the calculation in N-BEATS [25]. The following results demonstrate that our proposed Non-stationary Transformer can still improve upon the previous methods generally, which reduces 25.09% sMAPE on Transformer, 4.84% on Autoformer, and 5.60% on FEDformer on average. Note that the value of relative promotion can be changed under different metrics.
> >
> > |Exchange (sMAPE\|MASE)|Predict 96|Predict 192|Predict 336|Predict 720|
> > |-|-|-|-|-|
> > |Transformer|6.594 \| 22.948| 9.632 \| 32.386| 11.166 \| 36.892|14.252 \| 47.601|
> > |Transformer + Ours| 2.934 \| 9.435| 3.860 \| 12.692 | 5.650 \| 18.941|8.509 \| 29.881|
> > |Autoformer| 3.032 \| 10.076|4.194 \| 13.845|5.671 \| 18.744|8.904 \| 31.192|
> > |Autoformer + Ours|3.485 \| 11.633| 3.948 \| 13.372| 5.392 \| 18.100|9.483 \| 34.474|
> > |FEDformer|2.989 \| 9.818|4.271 \| 14.048|5.468 \| 18.205| 8.956 \| 31.602|
> > |FEDformer + Ours|2.857 \| 9.356|4.107 \| 13.451|5.578 \| 18.436| 9.528 \| 31.596|
> >
> > |ILI (sMAPE\|MASE)|Predict 24|Predict 36|Predict 48|Predict 60|
> > |-|-|-|-|-|
> > |Transformer|52.355 \| 7.231| 51.878 \| 7.260| 52.286 \| 8.171|51.998 \|  8.912|
> > |Transformer + Ours|35.391 \| 3.114| 30.770 \| 2.964| 30.485 \| 3.355|35.354 \| 3.685|
> > |Autoformer|47.399 \| 3.400|43.689 \| 3.919|39.585 \| 3.868| 40.946 \| 4.208|
> > |Autoformer + Ours|46.705 \| 3.377|43.416 \| 3.535|38.488 \| 3.760|39.003 \| 4.056|
> > |FEDformer|46.343 \| 3.381|37.618 \| 3.369| 37.906 \| 3.746|41.113 \| 4.157|
> > |FEDformer + Ours|42.662 \| 3.322| 34.048 \| 3.342| 34.716 \| 3.617| 37.777 \| 3.980|
> >
> > |ETTm2 (sMAPE\|MASE)|Predict 96|Predict 192|Predict 336|Predict 720|
> > |-|-|-|-|-|
> > |Transformer|72.888 \| 6.493| 86.689 \| 10.162| 95.832 \| 10.475 |97.860 \| 14.274|
> > |Transformer + Ours|61.897 \| 4.029| 65.713 \| 4.714 |70.782 \| 6.568|76.118 \| 7.548|
> > |Autoformer|62.317 \| 3.862|65.183 \| 4.111|67.377 \| 4.855|73.798 \| 5.141|
> > |Autoformer + Ours|61.434 \| 3.846| 65.232 \| 4.037| 67.157 \| 4.472| 71.554 \| 5.197|
> > |FEDformer|60.669 \| 3.392| 62.891 \| 3.910| 66.745 \| 4.408| 71.449 \| 5.077|
> > |FEDformer + Ours| 60.352 \| 3.328| 62.087 \| 3.906|68.259 \| 4.503| 72.295 \| 5.133|
> >
> > |Electricity (sMAPE\|MASE)|Predict 96|Predict 192|Predict 336|Predict 720|
> > |-|-|-|-|-|
> > |Transformer|17.646 \| 1.185|18.118 \| 1.318| 17.986 \| 1.246|18.599 \| 1.228|
> > |Transformer + Ours|14.068 \| 0.975|14.395 \| 1.018| 14.891 \| 1.068| 15.634 \| 1.092|
> > |Autoformer|17.066 \| 1.135|17.574 \| 1.216|18.958 \| 1.400| 19.569 \| 1.320|
> > |Autoformer + Ours|14.852 \| 1.034| 15.736 \| 1.109| 16.637 \| 1.186| 18.184 \| 1.304|
> > |FEDformer|17.273 \| 1.058| 17.405 \| 1.109| 18.277 \| 1.201| 18.479 \| 1.265|
> > |FEDformer + Ours|14.437 \| 0.999| 14.725 \| 1.051|15.549 \| 1.145|16.229 \| 1.170|
> >
> > |Traffic(sMAPE\|MASE)|Predict 96|Predict 192|Predict 336|Predict 720|
> > |-|-|-|-|-|
> > |Transformer|33.626 \| 1.056| 34.033 \| 1.033 | 33.173 \| 1.058|33.879 \| 1.139|
> > |Transformer + Ours|32.685 \| 0.987|33.189 \| 0.980| 32.441 \| 0.973| 33.371 \| 1.031|
> > |Autoformer|39.165 \| 1.139| 39.787 \| 1.202| 37.978 \| 1.099| 41.439 \| 1.234|
> > |Autoformer + Ours|36.182 \| 1.034| 38.019 \| 1.087| 36.384 \| 1.052| 37.858 \| 1.130|
> > |FEDformer|37.635 \| 1.072|37.921 \| 1.128|38.046 \| 1.125|38.674 \| 1.150
> > |FEDformer + Ours|34.639 \| 1.019| 35.206 \| 1.042|35.712 \| 1.054|36.678 \| 1.113
> >
> > (The weather benchmark is in the next part.)

---

> > > ### Author Response · Authors · 2022-08-01
> > > **Response to Reviewer q3Tk (Part 3)**
> > >
> > > Weather benchmark in Q2.
> > >
> > > |Weather (sMAPE\|MASE)|Predict 96|Predict 192|Predict 336|Predict 720|
> > > |-|-|-|-|-|
> > > |Transformer|73.607 \| 9.776|75.917 \| 9.987|75.185 \| 13.242|81.749 \| 12.850|
> > > |Transformer + Ours|60.462 \| 5.044| 63.586 \| 5.894| 64.736 \| 5.395| 68.112 \| 5.524|
> > > |Autoformer|68.726 \| 10.368| 69.467 \| 9.461|70.435 \| 10.666| 73.300 \| 10.921|
> > > |Autoformer + Ours|62.997 \| 6.733| 64.784 \| 6.394| 66.958 \| 6.204|  67.936 \| 4.987|
> > > |FEDformer|68.915 \| 11.076| 68.268 \| 9.736|71.029 \| 9.820| 76.752 \| 11.543|
> > > |FEDformer + Ours|60.699 \| 4.247|63.033 \| 4.421|65.047 \| 4.539| 66.958 \| 4.765|
> > >
> > > **Q3:** Add more advanced transform-based forecasting model for comparison.
> > >
> > > (1) Absolute comparison with ETSformer and FEDformer on MSE and MAE.
> > >
> > > As per your request, we add the ETSformer and FEDformer as the baselines in $\underline{\text{Table 2 of the main text}}$.
> > >
> > > It is notable that the FEDformer  (newly accepted by ICML 2022) and ETSformer (arXiv 2022) have not been officially published during our submission. Our method is comparable to this concurrent work. In addition, our method is a general framework, which can be applied to these advanced methods.
> > >
> > > |Exchange (MSE\|MAE)|Predict 96|Predict 192|Predict 336|Predict 720|
> > > |-|-|-|-|-|
> > > |Ours|0.111 \| 0.237|0.219 \| 0.335|0.421 \| 0.476|1.092 \| **0.769**|
> > > |ETSformer|**0.085** \| **0.204**| **0.182** \| **0.303**|**0.348** \| **0.428**|**1.025** \| 0.774|
> > > |FEDformer|0.148 \| 0.278|0.271 \| 0.380|0.460 \| 0.500|1.195 \| 0.841|
> > >
> > > |ILI (MSE\|MAE)|Predict 24|Predict 36|Predict 48|Predict 60|
> > > |-|-|-|-|-|
> > > |Ours|**2.294** \| **0.945**|**1.825** \| **0.848**|**2.010** \| **0.900**| **2.178** \| **0.963**|
> > > |ETSformer| 2.536 \| 1.021| 2.875 \| 1.082 | 2.536 \| 1.004| 2.529 \| 1.029|
> > > |FEDformer|3.228 \| 1.260|2.679 \| 1.080|2.622 \| 1.078|2.857 \| 1.157|
> > >
> > > |ETTm2 (MSE\|MAE)|Predict 96|Predict 192|Predict 336|Predict 720|
> > > |-|-|-|-|-|
> > > |Ours|0.192 \| **0.274**| 0.280 \| 0.339| 0.334 \| 0.361| 0.417 \| 0.413|
> > > |ETSformer|**0.189** \| 0.280| **0.253** \| **0.319**|**0.314** \| **0.357**|**0.414** \| **0.413**|
> > > |FEDformer|0.203 \| 0.287|0.269 \| 0.328| 0.325 \| 0.366|0.421 \| 0.415|
> > >
> > > |Electricity (MSE\|MAE)|Predict 96|Predict 192|Predict 336|Predict 720|
> > > |-|-|-|-|-|
> > > |Ours|**0.169** \| **0.273**| **0.182** \| **0.286**| **0.200**\| **0.304**| **0.222** \| **0.321**|
> > > |ETSformer|0.187 \| 0.304|0.199 \| 0.315|0.212 \| 0.329|0.233 \| 0.345|
> > > |FEDformer|0.193 \| 0.308|0.201 \| 0.315|0.214 \| 0.329|0.246 \| 0.355|
> > >
> > > |Traffic (MSE\|MAE)|Predict 96|Predict 192|Predict 336|Predict 720|
> > > |-|-|-|-|-|
> > > |Ours|0.612 \| **0.338** | 0.613 \| **0.340**| **0.618** \| **0.328** | 0.653 \| **0.355**|
> > > |ETSformer|0.616 \| 0.399| 0.645 \| 0.427| 0.628 \| 0.399|0.628 \| 0.388|
> > > |FEDformer|**0.587** \| 0.366|**0.604** \| 0.373|0.621 \| 0.383|**0.626** \| 0.382|
> > >
> > > |Weather (MSE\|MAE)|Predict 96|Predict 192|Predict 336|Predict 720|
> > > |-|-|-|-|-|
> > > |Ours|**0.173** \| **0.223**|0.245 \| **0.285**|0.321 \| **0.338**|0.414 \| 0.410|
> > > |ETSformer|0.197 \| 0.281|**0.237** \|0.312|**0.298** \| 0.353|**0.352**\| **0.388**|
> > > |FEDformer|0.217 \| 0.296|0.276 \| 0.336|0.339 \| 0.380|0.403 \| 0.428|
> > >
> > > (See the next part for the relative promotion.)

---

> > > > ### Author Response · Authors · 2022-08-01
> > > > **Response to Reviewer q3Tk (Part 4)**
> > > >
> > > > (2) Apply the Series Stationarization and the De-stationary Attention to ETSformer and FEDformer.
> > > >
> > > > Note that the Non-stationary Transformer can perform as a general framework and consistently promote the performance of various Transformers $\underline{\text{Table 4 of the main text}}$.
> > > >
> > > > To further address the effectiveness of our proposed Series Stationarization and the De-stationary Attention, we apply them with the advanced ETSformer and FEDformer. The experimental results demonstrate that our Non-stationary Transformer can further promote ETSformer and FEDformer.
> > > >
> > > > |Exchange (MSE\|MAE)|Predict 96|Predict 192|Predict 336|Predict 720|
> > > > |-|-|-|-|-|
> > > > |ETSformer|0.085 \| 0.204| 0.182 \| 0.303|0.348 \| 0.428|1.025 \| 0.774|
> > > > |ETSformer + Ours|**0.083** \| **0.201**|**0.177** \| **0.298**|**0.338** \| **0.420**|**0.878** \| **0.708**|
> > > > |FEDformer|0.148 \| 0.278|0.271 \| 0.380|0.460 \| 0.500|1.195 \| 0.841|
> > > > |FEDformer + Ours| **0.127** \| **0.254**| **0.251** \| **0.365**| **0.452** \| **0.497**| **1.168** \| **0.830**|
> > > >
> > > > |ILI (MSE\|MAE)|Predict 24|Predict 36|Predict 48|Predict 60|
> > > > |-|-|-|-|-|
> > > > |ETSformer|2.536 \| 1.021| 2.875 \| 1.082 | 2.536 \| **1.004**| 2.529 \| 1.029|
> > > > |ETSformer + Ours|**2.012** \| **1.005**|**2.518** \| **1.011**| **2.516** \| 1.030| **2.366** \| **1.022**|
> > > > |FEDformer|3.228 \| 1.260|2.679 \| 1.080|2.622 \| 1.078|2.857 \| 1.157|
> > > > |FEDformer + Ours|**3.200** \| **1.160**| **2.455** \| **0.969**|**2.484** \| **0.985**| **2.771** \| **1.069**|
> > > >
> > > >
> > > > |ETTm2 (MSE\|MAE)|Predict 96|Predict 192|Predict 336|Predict 720|
> > > > |-|-|-|-|-|
> > > > |ETSformer|0.189 \| 0.280| 0.253 \| 0.319|0.314 \| 0.357|0.414 \| 0.413|
> > > > |ETSformer + Ours|**0.187** \| **0.270**| **0.253** \| **0.310**|**0.312** \| **0.349**| **0.409** \| **0.405**|
> > > > |FEDformer|0.203 \| 0.287|0.269 \| 0.328| **0.325** \| **0.366**|**0.421** \| **0.415**|
> > > > |FEDformer + Ours|**0.191** \| **0.272**| **0.263** \| **0.317**| 0.343 \| 0.366| 0.450 \| 0.427|
> > > >
> > > > |Electricity (MSE\|MAE)|Predict 96|Predict 192|Predict 336|Predict 720|
> > > > |-|-|-|-|-|
> > > > |ETSformer|0.187 \| 0.304|0.199 \| 0.315|0.212 \| 0.329|0.233 \| 0.345|
> > > > |ETSformer + Ours|**0.177** \|  **0.289**| **0.193** \| **0.303**| **0.212** \| **0.321**| **0.231** \| **0.342**|
> > > > |FEDformer|0.193 \| 0.308|0.201 \| 0.315|0.214 \| 0.329|0.246 \| 0.355|
> > > > |FEDformer + Ours|**0.172** \| **0.278**|**0.184** \| **0.288**| **0.205** \| **0.310**| **0.230** \| **0.325**|
> > > >
> > > > |Traffic (MSE\|MAE)|Predict 96|Predict 192|Predict 336|Predict 720|
> > > > |-|-|-|-|-|
> > > > |ETSformer|0.616 \| 0.399| 0.645 \| 0.427| 0.628 \| 0.399|0.628 \| 0.388|
> > > > |ETSformer + Ours|**0.610** \| **0.370**| **0.614**\| **0.380**| **0.623** \| **0.386**| **0.626** \| **0.384**|
> > > > |FEDformer|0.587 \| 0.366|0.604 \| 0.373|0.621 \| 0.383|0.626 \| 0.382|
> > > > |FEDformer + Ours|**0.579** \| **0.348**| **0.599** \| **0.358**| **0.616** \| **0.363**| **0.623** \| **0.380**|
> > > >
> > > > |Weather (MSE\|MAE)|Predict 96|Predict 192|Predict 336|Predict 720|
> > > > |-|-|-|-|-|
> > > > |ETSformer|0.197 \| 0.281|0.237 \| 0.312|0.298 \| 0.353|0.352 \| 0.388|
> > > > |ETSformer + Ours|**0.169** \| **0.223**| **0.220**\| **0.269**| **0.284** \| **0.317**| **0.344** \| **0.362**|
> > > > |FEDformer|0.217 \| 0.296|0.276 \| 0.336|0.339 \| 0.380|0.403 \| 0.428|
> > > > |FEDformer + Ours|**0.187** \| **0.234**| **0.235** \| **0.271**| **0.289** \| **0.308**| **0.359** \| **0.353**|
> > > >
> > > >
> > > >
> > > > **Q4:** Hyper-parameter selection strategy for the baselines.
> > > >
> > > > (1) Most of the baselines are from the paper of Autoformer. By contacting the authors of Autoformer, we obtain the hyper-parameter selection strategy as follows:
> > > >
> > > > - N-BEATS: Grid search for hidden channel in {$256,512,768$}, number of layers in {$2,3,4,5$}, learning rate in {$5\times 10^{-5}, 1\times 10^{-4}, 5\times 10^{-4},1\times 10^{-3}$}.
> > > > - LSTNet: Since this paper also experiments on the Traffic, Electricity, and Exchange datasets, the hyper-parameter setting is following the experimental details of its own paper.
> > > >
> > > > (2) The hyper-parameter selection strategy of the new baselines, such as N-HiTs, ETSformer, and FEDformer. Since these methods share the same benchmark, we use their official code on GitHub with three random seeds.
> > > >
> > > > We have further clarified the hyper-parameter selection strategy in the supplementary materials of the $\underline{\text{revised paper}}$.

---

> ### Author Response · Authors · 2022-08-06
> **Discussion period only lefts 3 days**
>
> Dear Reviewer,
>
> We kindly remind you that it only left a few days of the one-week Reviewer-author discussion period. So please kindly let us know if our response has addressed your concerns. We will be happy to deal with any further issues/questions.
>
> We made every effort to address the concerns as you suggested:
>
> - We **verified the effectiveness of the normalization layer** from both literature and experimental aspects.
> - We reclarified the experiment setting, where **all the results are fairly compared in our original paper**.
> - We have **re-evaluated all main results with new metrics sMAPE and MASE**.
> - We added more advanced Transformers (ETSformer, FEDformer) as our baselines and further validated that **our design can further boost these advanced models**.
> - We have **included the hyper-parameter selection strategy** in the revised paper.
>
> Thanks again for your dedication to reviewing our paper.

---

> > ### Comment · Reviewer_q3Tk · 2022-08-09
> > **response to author feedback**
> >
> > Thanks authors for providing more experiments to show the effectiveness of the new normalization layer.
> >
> > However, I am not convinced for the statement "significant improvement" and "fair comparison" for this long sequence forecasting setting. Most of recent transformer based forecasting model claim significant improvement, with 50% on MSE. At the same time, you can also find some recent works [1,2,3]   use very simple architecture (even linear model) and time series decomposition strategy to achieve better performance than the transformer based models. I understand that the author directly report the results from early work, but I think existing works lack of fair comparison to shallow model, statistical model, or even non-learnable baselines. As shown in [4], a simple replication of historical data can achieve comparable performance to Informer. Then, the question is do we really need the complicated transformer based model for long sequence forecasting? Is it a waste of effort for the community?  Lastly, I understand that the authors follow the same experimental setting and report the normalized results, but it does not mean this setting is correct. From your results, we can also find the results would be quite different when using different evaluation metric.
> >
> > Aside from these questions and focus on the context of transformer-based model and non-stationary problem, I think the major concern of this work is the motivation is unclear. First, I think the authors should clarify the definition of "non-stationary" which indicates the distribution shift in both training and test data, not the traditional definition for stochastic process. This problem is interesting and important, which should be agnostic to the model architecture. However, I am unclear why the proposed method cannot be applied for other architecture, like MLP, RNN? Why it is transformer specific solution? Additionally, I am not clear of the formal definition of non-stationary, and how the proposed normalization is motivated from it? Is there any generalization guarantee for the proposed normalization in this setting with data-dependent distribution? What would happen if only the test data contain non-stationary change? The proposed analysis in Sec 3.2 seem to approximate the original self-attention with normalized data. Then the question is how to guarantee the original self-attention can tackle the non-stationary problem well?
> >
> > From the technique perspective, the proposed normalization layer seems to be heuristic. I am not clear about the justification of this design. I do not find the response to my question "why not remove the normalization before embedding layer and use the raw data for computation of attention score?".  It seems to be time series feature engineering and incorporate statistic features, like the mean, variance of the sequence. Why the architecture need to digest these features in this way? Besides, a straightforward extension is to use more statistic features, like the top solution in M4 competition [5]. Then, why do not consider other statistic features? Lastly, if the authors use the normalized data as the input, how to differentiate the effect of the data normalization and the layer normalization?
> >
> >
> > [1] Are Transformers Effective for Time Series Forecasting?
> > [2] DeepTIMe: Deep Time-Index Meta-Learning for Non-Stationary Time-Series Forecasting
> > [3] FreDo: Frequency Domain-based Long-Term Time Series Forecasting
> > [4] Historical Inertia: A Neglected but Powerful Baseline for Long Sequence Time-series Forecasting
> > [5] FFORMA: Feature-based Forecast Model Averaging

---

> > > ### Author Response · Authors · 2022-08-09
> > > **Response to the further questions of Reviewer q3Tk (Part 1)**
> > >
> > > Thanks for your response and questions.
> > >
> > > **Q1: “It does not mean this setting is correct.”**
> > >
> > > (1) All the works that you mentioned follow the same setting as our paper.
> > >
> > > The works [1,2,3,4,5] that the reviewer mentioned all follow the same setting as Informer, Autoformer, or our proposed Non-stationary Transformer. Thus, we think our setting is widely-used and a convention of this community.
> > >
> > > (2) Method comparison.
> > >
> > > It is also notable that, all the works that the reviewer mentioned are concurrent to our paper, such as DLinear [1] (26 May arXiv 2022), DeepTIMe [2] (13 Jul arXiv 2022), FreDo [3] (24 May 2022 arXiv 2022). And none of these methods are officially published.
> > >
> > > Besides, we have provided the N-BEATS (linear model), and ARIMA (classic method) as our baselines. And following the review's original review, we have compared the latest model FEDformer and ETSformer.
> > >
> > > (3) **Our model is state-of-the-art in Transformer-based models.**
> > >
> > > I think Transformer is a more powerful model than the pure linear model, which has been verified in extensive areas. And introducing powerful models into time series can benefit the future work of this community, such as designing big models or combining them with other modalities. And our model is state-of-the-art in this method paradigm.
> > >
> > > (4) We have provided a completely new benchmark by comparing methods in the original space.
> > >
> > > In the previous rebuttal, as the reviewer requested, we have provided the results in the original space. And in the original space, our Non-stationary Transformer can also outperform other baselines and bring further promotion.
> > >
> > > **Q2: "The motivation is unclear." "Why the proposed method cannot be applied to other architecture."**
> > >
> > > In this paper, we delve for the first time into **the concrete effect of "non-stationary time series" in Transformers**. The concrete effect is that the learned temporal dependencies (attention in Transformers) will be less distinguishable, which is defined as "over-stationarization". This finding motivates us to focus on the attention design and propose De-stationary Attention.
> > >
> > > For other architectures, such as MLP, RNN.
> > >
> > > - It is hard to visualize learned temporal dependencies, where we cannot find an explicit element to represent the temporal dependencies in these models.
> > > - The negative effect of "non-stationarity" in these models is under-explored.
> > > - All the theoretical analyses in our paper are based on self-attention. It is nontrivial to generalize these to MLP or RNNs.
> > >
> > > Thus, we will leave the exploration of the non-stationarity for other architectures in future work.
> > >
> > > **Q3: I don't find the response to "Why not remove the normalization before embedding layer and use the raw data for computation of attention score?"**
> > >
> > > Sorry for the unclearness. We have provided this in the previous rebuttal. Please see **Q1-(2)**:
> > >
> > > - $\underline{\text{only De-stationary Attention}}$ in the table means that "remove the normalization and use the raw data for computation of De-stationation attention".
> > > - $\underline{\text{vinalla Transformer}}$ in the table means that "remove the normalization and use the raw data for computation of vanilla self-attention".
> > >
> > > **Q4: "Why the architecture need to digest these features in this way?"**
> > >
> > > It is notable that by specifying the over-stationarization problem as the less distinguishable attention problem, we have narrowed down our design space into the attention calculation mechanism. Some other methods for the attention calculation of our model are included in the $\underline{\text{Table 9 of supplementary materials}}$, such as only $\tau$ and only $\mathbf{\Delta}$.
> > >
> > > To further address the review's concern, we also conduct an experiment by reincorporating $\mu$ and $\sigma$ into the feed-forward layer. But since the effect of stationarization on the feed-forward layer is under-explored, this "feed-forward" reincorporation design may not be as well-motivated as our De-stationary Attention. In most cases of the new experimental results (see table below), our proposed De-stationary Attention achieves better performance. Hence it is a more optimal design with theoretical support.
> > >
> > > | ETTm2 (MSE\|MAE)                  | Predict 96             | Predict 192            | Predict 336            | Predict 720            |
> > > | --------------------------------- | ---------------------- | ---------------------- | ---------------------- | ---------------------- |
> > > | Series Stationarization           | 0.253 \| 0.311         | 0.453 \| 0.404         | 0.546 \| 0.461         | 0.593 \| 0.489         |
> > > | + Reincorporation on Feed Forward | 0.275 \| 0.329         | 0.406 \| 0.403         | 0.502 \| 0.465         | 0.694 \| 0.575         |
> > > | + De-stationary Attention (Ours)  | **0.192** \| **0.274** | **0.280** \| **0.339** | **0.334** \| **0.361** | **0.417** \| **0.413** |
> > >
> > > The results on the full six benchmarks can be found in the response to Q3 of Reviewer PhZA.

---

> > > > ### Author Response · Authors · 2022-08-09
> > > > **Response to the further questions of Reviewer q3Tk (Part 2)**
> > > >
> > > > **Q6: "why do not consider other statistic features?"**
> > > >
> > > > **(1) Our proposed Series Stationarization is powerful enough in enhancing the time series stationarity.** The comparison of ADF test statistic is shown as follows. Note that a smaller value of ADF Test Statistic means more likely to be stationarity.
> > > >
> > > > | ADF test statistic            | Exchange   | ILI         | ETTm2       | Electricity | Traffic     |
> > > > | ----------------------------- | ---------- | ----------- | ----------- | ----------- | ----------- |
> > > > | Raw data                      | -1.889     | -5.406      | -6.225      | -8.483      | -15.046     |
> > > > | After Series Stationarization | **-9.937** | **-10.313** | **-33.485** | **-20.888** | **-18.946** |
> > > >
> > > > (2) The simple formulation in Series Stationarization can also benefit the derivation of De-stationary attention.
> > > >
> > > > (3) We would like to leave the more sophisticated normalization methods for Series Stationarization in future work, which is included in the conclusion of the revised paper.

---

### Official Review · Reviewer_PhZA · 2022-07-16

**Rating:** 4
**Confidence:** 2
**Soundness:** 2 fair
**Presentation:** 2 fair
**Contribution:** 2 fair

**Summary:**

The authors note that the stationarisation of time series removes information from the time series that can be used to aid in prediction. They call this problem the "over stationarisation of time series". Given that we want stationarity to improve model performance, but do not want to lose the information from the stationarisation transform, motivates the "Non-stationary transformer" introduced in the paper.

The model consists of a stationarisation component, and an additional component to reincorporate the stationarised information back into the time series.

They key component introduced is the De-stationary attention module. This transforms the dat a

**Questions:**

Why does the De-stationary attention module not just operate on the existing, non-stationarised features directly?

**Limitations:**

Yes

**Strengths And Weaknesses:**

It is certainly important to reincorporate the information removed by the stationarity transform into the prediction model. The proposed method introduces some method of doing this and demonstrates that it improves performance accross a wide range of benchmarks.

Stationarisation losing information and thus potential predictive power is intuitively obvious. The novelty here (to me at least) is the destationary attention mechanism. This is another attention block that has some scaling taking into account non-stationary information.

It's not clear to me why it has to be done in this way. Why can I not just incorporate mu sigma into the MLP applied to the output of the self attention? My biggest issue with the paper is the motivation behind it. Stationarity loses information - fine. Adding information back in some way should improve performance - fine. This is known and not novel. We add some additional transformer to the existing transformer that accounts for some of this information - fine, but I'm not sure why I should be so interested in this architecture? There are lots of ways we could add information back in, why should I choose this one?

The experiments do not compare other mechanisms that incorporate such information. I havent been given a reason to be particularly excited about this mechanism over some other method of sticking in mu and sigma. Is there some motivation behind this mechanism that I am missing?

It would be nice if you had compared with non transformer benchmarks, e.g. LSSL/GRU/

"We refine that the predictive capability of non-stationary series is essential in real-world forecasting. By detailed analysis, we find out that current stationarization approaches will lead to the over-stationarization problem, limiting the predictive capability of Transformers." - From what I can tell, this 'refinement' is a single image showing different attention weights in different scenarios. I dont think there is enough done on this point in this paper to define this as a contribution.


The writing is not great, I found many parts quite difficult to read. A couple of examples of difficult-to-parse sentences:

"However, non-stationarity is the constitutional property of real-world time series that can be entangled with essential temporal dependencies for forecasting"

"Considering the deep model scenario under the real case, de-stationary factors should be disentangled from the statistics of unstationarized x, Q and K."

In general I think a lot of the wording is overcomplicated and could be significantly simplified.

You say transformers perform well "credited to their stacked structure". Most DL models for time series are stacked in some way. I dont really see how this is a plus for transformers.

---

> ### Author Response · Authors · 2022-08-01
> **Response to Reviewer PhZA (Part 1)**
>
> We would like to sincerely thank Reviewer PhZA for providing the insightful review.
>
> **Q1:** Reclarify the contribution of the "over-stationarization" problem.
>
> The reviewer mentioned that "Stationarisation losing information and thus potential predictive power is intuitively obvious." We agree with this argument but also would like to highlight the status of the literature:
>
> - It is obvious that "stationarization loses information" but how does stationarization negatively influences model behaviors is more important for algorithm design and improvement. To our best knowledge, previous methods focus mainly on how to stationarize time series **without elaboration and mitigation of its negative effect**.
> - In this paper, we delve for the first time into **the concrete effect of "stationarization loses information" in Transformers**. We solve this problem by designing the De-stationary Attention in Transformers, which is new for time series forecasting.
>
> Concretely, with insights analysis, we find out that a direct stationarization will cause **less distinguishable attention (temporal dependencies)** among different time series, which is defined as the over-stationarization problem ($\underline{\text{lines 40-42 of the main text}}$). We further clarify the following three items.
>
> - We focus on time series forecasting, in which the learned temporal dependencies (attention in Transformers) are essential to the forecasting performance. The over-stationarization problem causing less distinguishable attention is closely related to time series forecasting with Transformers.
> - Our found over-stationarization problem refers to a concrete situation of the degenerated model-learned attentions in Transformers. This finding is well-supported by the visualization in $\underline{\text{Figure 1 of the main text}}$ and the statistics in $\underline{\text{Figure 4 of the main text}}$.
> - The particular form of over-stationarization specifies our design space to be avoiding the less distinguishable attention in Transformers. Mitigating the over-stationarization in this specific design space leads us to the Non-stationary Transformer.
>
>
> **Q2:** Why incorporate $\mu$ and $\sigma$ this way? Clarify the motivation of De-stationary Attention design.
>
> As clarified in **Q1**, our design is based on the findings of the over-stationarization problem in Transformers. The particular design is also directly motivated by the derivation of the vanilla attention (plain model) over non-stationary time series.
>
> For a clear clarification, we sum up the motivations as the following pipeline:
>
> | Motivation                                                   | Design                                                       |
> | ------------------------------------------------------------ | ------------------------------------------------------------ |
> | Stationarization is important to time series forecasting ($\underline{\text{lines 28-36 of the main text}}$). | We adopt the Series Stationarization to enhance the stationarity of the input series. |
> | (1) Directly stationarization will cause less distinguishable attention in Transformers (over-stationarization problem). (2) The attention corresponds to the learned temporal dependencies, and therefore the less distinguishable attention will affect the forecasting performance. | **We focus on the attention calculation mechanism and attempt to avoid the less distinguishable attention**, namely avoiding the over-stationarization problem ($\underline{\text{lines 40-42 of the main text}}$). |
> | (1) Transformer can discover the particular temporal dependencies from raw series. (2) The input series is stationarized now. | It is a natural and direct way to **approximate the particular attention learned by Transformer without stationarization** ($\underline{\text{line 147 of the main text}}$). |
> | The analysis and derivation of the plain model in $\underline{\text{Section 1 of supplementary materials}}$. | We can reincorporate $\mu$ and $\sigma$ to the less distinguishable attention map $\text{Softmax}(\frac{\mathbf{Q}^\prime{\mathbf{K}^\prime}^\top}{\sqrt{d_k}})$ as $\underline{\text{Equation 6 of the main text}}$ to **approximate the desired attention** $\text{Softmax}(\frac{\mathbf{Q}\mathbf{K}^\top}{\sqrt{d_k}})$ learned from raw data. |
>
> Motivated by the above insights and derivations, we design the De-stationary Attention as a direct complement of the Series Stationarization to avoid the over-stationarization problem.

---

> > ### Author Response · Authors · 2022-08-01
> > **Response to Reviewer PhZA (Part 2)**
> >
> > **Q3:** Other ways to reincorporate the non-stationary information.
> >
> > To our best knowledge, this is the first work that explores the co-design of stationarization and de-stationarization. Thus from previous papers, we cannot obtain other ideas for other designs to "reincorporate the non-stationary information".
> >
> > - It is notable that by specifying the over-stationarization problem as the less distinguishable attention problem, we have narrowed down our design space into the attention calculation mechanism. Some other methods for the attention calculation of our model are included in the $\underline{\text{Table 9 of supplementary materials}}$, such as only $\tau$ and only $\mathbf{\Delta}$.
> >
> > - To further address the review's concern, we also conduct an experiment by reincorporating $\mu$ and $\sigma$ into the feed-forward layer as the reviewer suggested. But since the effect of stationarization on the feed-forward layer is under-explored, this "feed-forward" reincorporation design may not be as well-motivated as our De-stationary Attention. In most cases (83%) of the new experimental results (see table below), our proposed De-stationary Attention achieves better performance. Hence it is a more optimal design with theoretical support.
> > - Our literature survey contradicts the comment that "there are lots of ways we could add information back in". It will be very helpful if the reviewer could give some citations/references for other possible designs.
> >
> > |Exchange (MSE\|MAE)|Predict 96|Predict 192|Predict 336|Predict 720|
> > |-|-|-|-|-|
> > |Series Stationarization|0.136 \| 0.258|0.239 \| 0.348|0.425 \| 0.479|1.475 \| 0.865|
> > |+ Reincorporation on Feed Forward|0.116 \| 0.243|0.280 \| 0.383|**0.371** \| **0.452**|**0.634** \| **0.604**|
> > | + De-stationary Attention (Ours)|**0.111** \| **0.237**|**0.219** \| **0.335**|0.421 \| 0.476|1.092 \| 0.769|
> >
> > |ILI (MSE\|MAE)|Predict 24|Predict 36|Predict 48|Predict 60|
> > |-|-|-|-|-|
> > |Series Stationarization|2.573 \| 0.980 | 1.955 \| 0.870|2.057 \| 0.902|2.238 \| 0.982|
> > |+ Reincorporation on Feed Forward|2.404 \| 0.985|2.585 \| 0.983|2.496 \| 0.991 | 2.667 \| 1.059|
> > |+ De-stationary Attention (Ours)|**2.294** \| **0.945**|**1.825** \| **0.848**|**2.010** \| **0.900**| **2.178** \| **0.963**|
> >
> > |ETTm2 (MSE\|MAE)|Predict 96|Predict 192|Predict 336|Predict 720|
> > |-|-|-|-|-|
> > |Series Stationarization|0.253 \| 0.311| 0.453 \| 0.404|0.546 \| 0.461|0.593 \| 0.489|
> > |+ Reincorporation on Feed Forward|0.275 \| 0.329| 0.406 \| 0.403|0.502 \| 0.465|0.694 \| 0.575|
> > |+ De-stationary Attention (Ours)|**0.192** \| **0.274**| **0.280** \| **0.339**| **0.334** \| **0.361**| **0.417** \| **0.413**|
> >
> > |Electricity (MSE\|MAE)|Predict 96|Predict 192|Predict 336|Predict 720|
> > |-|-|-|-|-|
> > |Series Stationarization|0.171 \| 0.275| 0.192 \| 0.296| 0.208 \| 0.306| **0.216** \|  **0.315**|
> > |+ Reincorporation on Feed Forward| 0.170 \| 0.274| 0.188 \| 0.293| 0.206 \| 0.309|0.223 \| 0.323|
> > |+ De-stationary Attention (Ours)|**0.169** \| **0.273**| **0.182** \| **0.286**| **0.200** \| **0.304**| 0.222 \| 0.321|
> >
> > |Traffic (MSE\|MAE)|Predict 96|Predict 192|Predict 336|Predict 720|
> > |-|-|-|-|-|
> > |Series Stationarization|0.614 \| 0.337|0.637 \| 0.351| 0.653 \| 0.359| 0.661 \| 0.360|
> > |+ Reincorporation on Feed Forward|**0.605** \| **0.333**| 0.617 \| 0.342| 0.635 \| 0.349| **0.649** \| **0.351**|
> > |+ De-stationary Attention (Ours)|0.612 \| 0.338 | **0.613** \| **0.340**| **0.618** \| **0.328** | 0.653 \| 0.355|
> >
> > |Weather (MSE\|MAE)|Predict 96|Predict 192|Predict 336|Predict 720|
> > |-|-|-|-|-|
> > |Series Stationarization|0.175 \| 0.225| 0.273 \| 0.297| 0.333 \| **0.325**| 0.436 \| 0.420|
> > |+ Reincorporation on Feed Forward|0.178 \| 0.226|0.256 \| 0.295|0.338 \| 0.351 |0.417 \| 0.412|
> > |+ De-stationary Attention (Ours)|**0.173** \| **0.223**|**0.245** \| **0.285**|**0.321** \| 0.338|**0.414** \| **0.410**|

---

> > > ### Author Response · Authors · 2022-08-01
> > > **Response to Reviewer PhZA (Part 3)**
> > >
> > > **Q4:** Why does De-stationary Attention not just operate on non-stationarized features directly?
> > >
> > > (1) Literature analysis.
> > >
> > > Note that the non-stationarized features can only be obtained when the input is not stationarized. But the non-stationary time series is hard for forecasting. As we stated in the $\underline{\text{lines 29-34 of the main text}}$, the non-stationary time series can affect the prediction in both data and deep learning views.
> > >
> > > - "The non-stationary time series is less predictable."
> > > - "Non-stationary time series corresponds to the change of statistical properties and joint distributions over time. It is a fundamental but challenging problem to make deep models generalize well on a varying distribution."
> > >
> > > Based on the above considerations, we propose Non-stationary Transformer with the interdependent Series Stationarization for improving the series predictability and De-stationary Attention for recovering the nonstationary information.
> > >
> > > (2) Experimental results.
> > >
> > > The benefits of Series Stationarization and De-stationary Attention have been verified in the ablation study ($\underline{\text{Table 5 of main text}}$). As per your request, we newly add an ablation study by removing the Series Stationarization and only maintaining the De-stationary Attention. Here are the results. We have the following two observations:
> > >
> > > - "Only De-stationary Attention" can surpass the Vanilla Transformer in datasets with stronger stationarity, such as ETTm2, Electricity, Traffic, and Weather.
> > > - Series-stationarization can bring benefits to all benchmarks. For the datasets with stronger non-stationarity, the benefits of Series-stationarization can be more significant.
> > >
> > > In summary, Series Stationarization and De-stationary Attention work interdependently. Both designs are necessary.
> > >
> > > |Exchange (MSE\|MAE)|Predict 96|Predict 192|Predict 336|Predict 720|
> > > |-|-|-|-|-|
> > > |Vanilla Transformer|0.567 \| 0.591 | 1.150 \| 0.825 |1.792 \| 1.084 | 2.191 \| 1.159 |
> > > | + only De-stationary Attention| 0.611 \| 0.613|1.202 \| 0.840|1.516 \| 0.981 | 2.894 \| 1.377|
> > > | + Ours|**0.111** \| **0.237**|**0.219** \| **0.335**|**0.421** \| **0.476**|**1.092** \| **0.769**|
> > >
> > > |ILI (MSE\|MAE)|Predict 24|Predict 36|Predict 48|Predict 60|
> > > |-|-|-|-|-|
> > > |Vanilla Transformer|4.748 \| 1.430|4.671 \| 1.430|4.994 \| 1.482|5.041 \| 1.499|
> > > | + only De-stationary Attention|4.734 \| 1.424|4.927 \| 1.482| 4.996 \| 1.483|5.184 \| 1.519|
> > > | + Ours|**2.294** \| **0.945**|**1.825** \| **0.848**|**2.010** \| **0.900**| **2.178** \| **0.963**|
> > >
> > > |ETTm2 (MSE\|MAE)|Predict 96|Predict 192|Predict 336|Predict 720|
> > > |-|-|-|-|-|
> > > |Vanilla Transformer|0.572 \| 0.552| 1.161 \| 0.793|1.209 \| 0.842| 3.061 \| 1.289|
> > > | + only De-stationary Attention|0.304 \| 0.406| 0.820 \| 0.652| 1.406 \| 0.883| 2.858 \| 1.108|
> > > | + Ours|**0.192** \| **0.274**| **0.280** \| **0.339**| **0.334** \| **0.361**| **0.417** \| **0.413**|
> > >
> > > |Electricity (MSE\|MAE)|Predict 96|Predict 192|Predict 336|Predict 720|
> > > |-|-|-|-|-|
> > > |Vanilla Transformer|0.260 \| 0.358| 0.266 \| 0.367| 0.280 \| 0.375|0.302 \| 0.386|
> > > | + only De-stationary Attention|0.253 \| 0.351|0.257 \| 0.358| 0.270 \| 0.365 |0.295 \| 0.380|
> > > | + Ours|**0.169** \| **0.273**| **0.182** \| **0.286**| **0.200** \| **0.304**| **0.222** \| **0.321**|
> > >
> > > |Traffic (MSE\|MAE)|Predict 96|Predict 192|Predict 336|Predict 720|
> > > |-|-|-|-|-|
> > > |Vanilla Transformer|0.647 \| 0.357|0.649 \| 0.356|0.667 \| 0.364| 0.697 \| 0.376|
> > > | + only De-stationary Attention|0.650 \| 0.358| 0.655 \| 0.358| 0.656 \| 0.355|0.681 \| 0.366|
> > > | + Ours|**0.612** \| **0.338** | **0.613** \| **0.340**| **0.618** \| **0.328** | **0.653** \| **0.355**|
> > >
> > > |Weather (MSE\|MAE)|Predict 96|Predict 192|Predict 336|Predict 720|
> > > |-|-|-|-|-|
> > > |Vanilla Transformer| 0.395 \| 0.427 | 0.619 \| 0.560 | 0.689 \| 0.594 | 0.926 \| 0.710 |
> > > | + only De-stationary Attention| 0.296 \| 0.364| 0.480 \| 0.464 | 0.581 \| 0.519 | 0.795 \| 0.642|
> > > | + Ours|**0.173** \| **0.223**|**0.245** \| **0.285**|**0.321** \| **0.338**|**0.414** \| **0.410**|

---

> > > > ### Author Response · Authors · 2022-08-01
> > > > **Response to Reviewer PhZA (Part 4)**
> > > >
> > > > **Q5:** Compare with other benchmarks
> > > >
> > > > As shown in the $\underline{\text{Table 2 and Table 3 of main text}}$, except Transformers, we have compared with various baselines, including the linear models: N-Beats (2019) and N-HiTs (2022), the LSTM-based baselines: LSTNet (2018) and the Classical method: ARIMA.
> > > >
> > > > As per your request, we also include the GRU and LSSL as our baselines. Here are the results. The proposed Non-stationary Transformer still achieves the best performance in all benchmarks. Notably, LSSL can give a good performance on the Weather dataset with the strongest stationarity, but fails in other datasets, especially the datasets with strong non-stationarity.
> > > >
> > > > |Exchange (MSE\|MAE)|Predict 96|Predict 192|Predict 336|Predict 720|
> > > > |-|-|-|-|-|
> > > > |GRU|1.453 \| 1.049| 1.846 \| 1.179| 2.136 \| 1.231| 2.984 \| 1.427|
> > > > |LSSL|0.395 \| 0.474|0.776 \| 0.698| 1.029 \| 0.797|2.283 \| 1.222|
> > > > |Ours|**0.111** \| **0.237**|**0.219** \| **0.335**|**0.421** \| **0.476**|**1.092** \| **0.769**|
> > > >
> > > > |ILI (MSE\|MAE)|Predict 24|Predict 36|Predict 48|Predict 60|
> > > > |-|-|-|-|-|
> > > > |GRU|5.914 \| 1.734| 6.631 \| 1.845| 6.736 \| 1.857| 6.870 \| 1.879|
> > > > |LSSL|4.381 \| 1.425|4.442 \| 1.416|4.559 \| 1.443|4.651 \| 1.474|
> > > > |Ours|**2.294** \| **0.945**|**1.825** \| **0.848**|**2.010** \| **0.900**| **2.178** \| **0.963**|
> > > >
> > > > |ETTm2 (MSE\|MAE)|Predict 96|Predict 192|Predict 336|Predict 720|
> > > > |-|-|-|-|-|
> > > > |GRU|2.041 \| 1.073 | 2.249 \| 1.112| 2.568 \| 1.238| 2.720 \| 1.287|
> > > > |LSSL|0.243 \| 0.342|0.392 \| 0.448|0.932 \| 0.724|1.372 \| 0.879|
> > > > |Ours|**0.192** \| **0.274**| **0.280** \| **0.339**| **0.334** \| **0.361**| **0.417** \| **0.413**|
> > > >
> > > > |Electricity (MSE\|MAE)|Predict 96|Predict 192|Predict 336|Predict 720|
> > > > |-|-|-|-|-|
> > > > |GRU|0.375 \| 0.437|0.442 \| 0.473| 0.439 \| 0.473|0.980 \| 0.814|
> > > > |LSSL|0.300 \| 0.392|0.297 \| 0.390|0.317 \| 0.403|0.338 \| 0.417|
> > > > |Ours|**0.169** \| **0.273**| **0.182** \| **0.286**| **0.200** \| **0.304**| **0.222** \| **0.321**|
> > > >
> > > > |Traffic (MSE\|MAE)|Predict 96|Predict 192|Predict 336|Predict 720|
> > > > |-|-|-|-|-|
> > > > |GRU|0.843 \| 0.453|0.847 \| 0.453|0.853 \| 0.455| 1.500 \| 0.805|
> > > > |LSSL|0.798 \| 0.436|0.849 \| 0.481|0.828 \| 0.476|0.854 \| 0.489|
> > > > |Ours|**0.612** \| **0.338** | **0.613** \| **0.340**| **0.618** \| **0.328** | **0.653** \| **0.355**|
> > > >
> > > > |Weather (MSE\|MAE)|Predict 96|Predict 192|Predict 336|Predict 720|
> > > > |-|-|-|-|-|
> > > > |GRU|0.369 \| 0.406| 0.416 \| 0.435|0.455 \| 0.454| 0.535 \| 0.520|
> > > > |LSSL|0.174 \| 0.252|**0.238** \| 0.313|**0.287** \| 0.355|**0.384** \| 0.415|
> > > > |Ours|**0.173** \| **0.223**|0.245 \| **0.285**|0.321 \| **0.338**|0.414 \| **0.410**|
> > > >
> > > >
> > > > **Q6:** The writing issues.
> > > >
> > > > - Simplify the writing: Thanks for your valuable suggestions. We have rephrased all the long sentences and overcomplicated words. All the changes are included in the updated $\underline{\text{revised paper}}$.
> > > > - The description of "credited to their stacked structure": In the original paper, the full description is "credited to their stacked structure and the capability of Self-Attention", which means that the Self-Attention can capture the dependencies from deep multi-level features. To eliminate the misunderstanding, we have rephrased this to "credited to their stacked structure and the capability of Self-Attention, Transformers can naturally capture the temporal dependencies from deep multi-level features" in the $\underline{\text{revised paper}}$.

---

> ### Author Response · Authors · 2022-08-06
> **Discussion period only lefts 3 days**
>
> Dear Reviewer,
>
> We kindly remind you that it has been 4 days since the one-week Reviewer-author discussion began. So please kindly let us know if our response has addressed your concerns. We will be happy to deal with any additional issues/questions.
>
> Following your suggestion, we revised the paper in the following aspects:
>
> - We reclarified our motivation and **conducted as many experiments as we can to verify the design of De-stationary Attention**.
> - We add **two new baselines (LSSL and GRU)** and evaluate them on all benchmarks to validate the performance of our model.
> - We rephrased the paper to solve every detailed confusing issue and **simplified all the overcomplicated descriptions**.
>
> Thanks again for your valuable review. Looking forward to your reply.

---

### Author Response · Authors · 2022-08-02
**Summary of Revisions**

We sincerely thank all the reviewers for their insightful reviews and valuable comments, which are instructive for us to improve our paper further.

In this paper, we explore time series forecasting from the perspective of stationarity. Unlike previous works that solely focus on the stationarization method itself, we are the first to notice the negative effect of direct stationarization and specify the "over-stationarization" problem. Based on the plain model analysis, we propose the Non-stationary Transformers as a general framework, which can improve the data predictability and avoid the over-stationarization problem simultaneously. Experimentally, our method consistently boosts Transformer and its variants remarkably (over 40% MSE reduction).

The reviewers generally held positive opinions of our paper, in that **our motivation “is clear”**, our writing **“is easy to follow”**and **“well organized”**, our experimental results show **“impressive” and “significantly improvement”**, our model **“is nice and elegant”** and our paper **“really brings great things to the table”**.

The reviewers also raised insightful and constructive concerns. We made every effort to address all the concerns by providing sufficient evidence and requested results. See the $\underline{\text{revised paper}}$ and $\underline{\text{supplementary materials}}$ for details. All updates are highlighted in blue and here is the summary of the revisions:

* **Motivations (Reviewers PhZA):** We highlight the contribution of noticing the negative effect of stationarization and specifying it as the concrete over-stationarization problem. And by emphasizing the instruction of theoretical analysis and evaluating every possible design of De-stationary Attention, we illustrate our motivation in both theoretical and experimental aspects.
* **Baselines and new metrics (Reviewers PhZA, q3Tk):** We enlarge our comparison from 9 to 13 baselines on all six benchmarks, covering the advanced recurrent models and concurrent works. Besides, we add the sMAPE and MASE as new metrics and re-evaluate all the main results. By doing great efforts to complete these comparisons, we verify that our method still achieves the best performance and good generality on new baselines and new metrics.
* **Ablation study of model design (Reviewers PhZA, q3Tk, 8mJi):** We added a comprehensive ablation, including the effect and design of Series stationarization (replenish the Table 5 of main text), design space of De-stationary Attention (replenish the Table 9 of supplement). We believe that the revised paper covers every detailed ablation of the motivation, designs, performance, and insight analysis.
* **Theoretical part statement (Reviewer ZUaG):** We reformulate some equations for clearness, remove unnecessary Gaussian distribution assumptions in the plain model analysis, and correct the false statement in distribution matching.
* **Concept definition (Reviewer ZUaG):** For scientific rigor, we have updated all the usages of stationarity concepts in the revised paper. We rephrase the “stationarity” to “the degree of stationarity”, which is a measurement of how stable the data distribution is. And we further define “Stationarization/De-stationarization” as a method to change the degree of stationarity.


The valuable suggestions from reviewers are very helpful for us to revise the paper to a better shape. We'd be very happy to answer any further questions.

---

### Meta-Review · Area_Chair_SPc8 · 2022-08-26

**Recommendation:** Accept
**Confidence:** Certain

**Metareview:**

The paper introduces a transformer-based method for non-stationary time series forecasting.
This research addresses a clear need, as acknowledged by the reviewers. Also, most reviewers found the method clearly described and the experiments compelling, demonstrating an improvement of the state of the art.

The reviewers asked questions about the baselines, evaluation methods and ablation studies. They also made requests related to clarifying the wording and some of the theory. The authors put in significant effort in addressing the comments, offering detailed responses to every reviewer. Only one of the reviewers responded during the discussion period, and the response came very late in the discussion period. However, I read the authors' response and concluded that they adequately addressed most issues raised by the reviewers.

As the model is in the Transformer space, and transformers have previously been shown to be state of the art on a number of tasks, I do not find it necessary to compare against other 'families' of methods. So I will consider that issue addressed as well.

**Award:**

No

---

### Decision · Program_Chairs · 2022-09-14

Accept